# SnapWave: fast, implicit wave transformation from offshore to nearshore

Dano Roelvink[1,2,3], Maarten van Ormondt[4], Johan Reyns[1,2,3], Marlies van der Lugt[2,3]

[1]Coastal and Urban Risk & Resilience, IHE Delft Institute for Water Education, Delft, 2611AX, the Netherlands
[2]Marine and Coastal Systems, Deltares, Delft, 2629 HV, the Netherlands
[3] Civil Engineering and Geosciences, Delft University of Technology, Delft, 12628 CN, the Netherlands
[4] Deltares USA Inc, Silver Spring, MD20910, United States of America

*Correspondence to*: Dano Roelvink (d.roelvink@un-ihe.org)

## Abstract

This paper presents an efficient, implicit, unstructured-grid wave propagation model, SnapWave (Roelvink et al., 2025), which provides a simple and fast way to predict nearshore wave conditions at specified locations, for coastline models such as ShorelineS, or wave fields and their forcing of flows, to be used in other models, such as Delft3D-FM, XBeach or SFINCS. We describe the numerical method and verify the correct implementation by comparing against analytical solutions for schematized cases. We then test the model application in four different coastal settings by propagating time series of ERA5
hourly wave conditions to observation points nearshore and through the surf zone. We conclude that the model is robust, easy to set up and fast, and can be applied on open coasts worldwide.

## 1 Introduction

The simulation of wave propagation and dissipation in coastal areas is important to transform wave fields from offshore areas where wave conditions are available from wave buoys or large-scale wave models to conditions nearshore. The nearshore
bathymetry controls the alongshore distribution of wave heights and directions and thereby, to a large extent, the shape and orientation of coastlines. Wave energy dissipation by bottom friction is important when waves pass over large shallow areas, whereas wave breaking dominates the distribution of wave energy through nearshore areas.

One of the earliest grid-based models for nearshore wave propagation and dissipation was HISWA (Holthuijsen et al., 1989), which applied a forward-marching technique on rectangular grids and applied a parameterized frequency spectrum while
resolving the directional spectrum. It was fast and reasonably accurate in nearshore areas, but had some disadvantages, such as that the wave grids had to be turned in the wave direction because of the forward-marching technique, and it lacked the full spectral description in frequency and direction. HISWA's successor, SWAN (Booij et al., 1999; Ris et al., 1999), is fully spectral and had third-generation wind growth terms similar to ocean wave models such as WAM (Wamdi-Group, 1988), WAVEWATCH (Tolman, 1991) and WAVEWATCH III (Tolman et al., 2002). The SWAN model runs on curvilinear grids,
which made it relatively easy to couple with morphological models such as Delft3D (Lesser et al., 2004). Another much-

applied stationary wave model, STWAVE (Smith, 2001), can be applied both in half-plane (forward-marching) or full-plane (all directions) mode; it is fully spectral, with simplified wind-wave growth formulations, and operates on a regular grid.

Although curvilinear meshes allow some flexibility in providing resolution where needed, this is limited because the meshes are structured, 2D in space. Unstructured meshes, whether triangular or a combination of quadrilaterals and triangles, do not have this constraint and can have local refinements that do not radiate out. Several models have been developed using such unstructured meshes: TOMAWAC (Benoit et al., 1997), unSWAN (Zijlema, 2010), WWM (Roland et al., 2012). While these models run much more efficiently thanks to the efficient distribution of grid resolution, see e.g. (Alves et al., 2022), the fact that they fully resolve the evolution of the 2D wave spectrum, and apply complex four-wave and triad interactions, still makes them quite computationally expensive when applied over larger areas with high resolution in space and time. To overcome such constraints (O'reilly et al., 2016) designed a characteristics-based method to link wave conditions outside the surf zone to a network of observation buoys along the California coast. This method resolved wave refraction, shoaling and sheltering of all components of the 2D spectrum, considering the travel time between the offshore buoys and the nearshore, while neglecting wave growth and dissipation, and provided reliable and fairly accurate predictions of nearshore wave conditions in most of the target locations.

For the modelling of nearshore morphological changes and for coastal inundation modelling, the wave model, inside a system or coupled with it, is usually the most time-consuming component. This is mostly due to processes, such as fully spectral modelling of non-linear interactions, that are only of secondary importance in nearshore areas. Even wave growth by wind can often be neglected on open coasts, when the waves only need to travel over tens of kilometres after having been generated over hundreds to thousands of kilometres. Therefore, we aim for a fast, stationary solver capturing only the essential physics of wave refraction and shoaling, dissipation by bottom friction and wave breaking. This solver, called SnapWave, presently serves the following purposes:

- an unstructured solver to resolve wave conditions along a nearshore depth contour, for coastline modelling in ShorelineS (Roelvink et al., 2020)
- an improved stationary wave solver for XBeach (Roelvink et al., 2009), allowing wave propagation in all directions;
- a stationary wave solver for unstructured grids consisting of triangular and quadrangular cells in Delft3D-FM (Reyns et al., 2023);
- a fast nearshore wave solver coupled with SFINCS, to resolve wave setup in inundation modelling (Leijnse et al., 2024).

In this paper, we present the first stage of this model, suitable for most open coasts. The main point of the paper is to demonstrate the use of SnapWave to efficiently transform wave conditions provided by a global model such as ERA5 (Hersbach et al., 2020) to nearshore locations anywhere in the world. After describing the numerical method, we verify the model implementation by comparing it with an analytical solution for refraction and shoaling on a straight coastline, and by analysing the iteration process for the case of a circular island and a circular reef. We then proceed with field cases of varying complexity, followed by a discussion and conclusions.

## 2. Model description

### 2.1 Coupled wave action balance and wave energy balance

We solve the wave action balance:

$$\frac{\partial aa}{\partial t} + \frac{\partial aaC_g \cos \vartheta}{\partial x} + \frac{\partial aaC_g \sin \vartheta}{\partial y} + \frac{\partial aaC_\vartheta}{\partial \vartheta} = ss_A - dd_A \quad (1)$$

With $aa$ the frequency-integrated, directionally distributed wave action density:

$$aa = \frac{ee}{\sigma} \qquad (2)$$

where $ee$ is the directionally distributed wave energy density and $\sigma$ is the relative angular frequency. $C_g$ the group velocity, $\vartheta$ the wave direction, $ss_A$ the wind source term and $dd_A$ the directionally distributed dissipation. In case the wind-growth source term is included in the balance, the wave period cannot be assumed spatially uniform over the domain and its evolution over the interior of the domain needs to be modelled as well. This is done by simultaneously solving the action balance and the wave energy balance:

$$\frac{\partial ee}{\partial t} + \frac{\partial ee\,C_g \cos \vartheta}{\partial x} + \frac{\partial ee\,C_g \sin \vartheta}{\partial y} + \frac{\partial ee\,C_\vartheta}{\partial \vartheta} = ss_E - dd \qquad (3)$$

where $dd$ is the wave dissipation density and $ss_E$ the wind source term. The representative frequency is then calculated as:

$$\sigma = \frac{E}{A} \qquad (4)$$

where

$$A = \int_0^{2\pi} aa\,d\vartheta, \quad E = \int_0^{2\pi} ee\,d\vartheta, \qquad (5)$$

Similar to HISWA and XBeach, we apply a parameterized frequency spectrum represented by a single frequency close to the peak frequency. The directional spectrum is resolved with a given, constant directional resolution and directional sector.

### 2.2 Simplification to wave energy balance

The main purpose of this paper is to present the numerical method of SnapWave and its application to propagating wave conditions from deep water to nearshore areas over relatively short distances. In many cases the effect of wave growth by wind and ambient currents is small, and here we focus on such cases. In a forthcoming paper we will detail the method of including

90 wave growth by wind, which is relevant in lakes, estuaries and tidal inlets where the wave climate is dominated by local wind waves.

Under the assumption of no wave growth by wind and neglecting ambient currents, the wave frequency is constant over the domain and the wave action balance (1) reduces to the wave energy balance (3), and the wind input term $ss_E$ goes to zero.

We assume that the directional distribution of the dissipation density $dd$ is the same s the distribution of the wave energy density $ee$, so:

$$dd = \frac{D}{E} ee \qquad (6)$$

Here, the total dissipation $D$ is the sum of the dissipation by wave breaking, $D_w$ and by friction, $D_f$.

$$D = D_w + D_f \qquad (7)$$

The wave breaking dissipation integrated over the directional spectrum, $D_w$ is according to Baldock et al. (1998):

$$D_w = 0.25 \alpha \rho g f_p \exp\left(-\frac{H_{max}^2}{H_{rms}^2}\right)\left(H_{max}^2 + H_{rms}^2\right) =$$
$$= 2\alpha f_p \exp\left(-\frac{E_{max}}{E}\right)\left(E_{max} + E\right) \qquad (8)$$

Here, $\alpha$ is a dissipation coefficient of order 1, $\rho$ the density of water, $f_p$ the peak frequency, $E$ the wave energy integrated over the directional spectrum, $E = 1/8\rho g H_{rms}^2$, $H_{rms}$ the root-mean-square wave height, $E_{max} = 1/8\rho g H_{max}^2$ and $H_{max}$ a depth- and

105 frequency-dependent maximum wave height given by:

$$H_{max} = \frac{0.88}{k} \tanh(\frac{\gamma kh}{0.88}) \qquad (9)$$

The dissipation by bottom friction is given by (Collins, 1972) :

$$D_w = 0.28 \rho f_w u_{rms}^3, \quad u_{rms} = \frac{\omega H_{rms}}{2 \sinh(kh)} \qquad (10)$$

We can write equation (3) in simpler form if we consider $s$ to be the distance along each wave direction:

$$\frac{\partial ee}{\partial t} + \frac{\partial ee\, C_g}{\partial s} + \frac{\partial ee\, C_\vartheta}{\partial \vartheta} + dd = 0 \qquad (11)$$

In the absence of currents, the refraction speed $C_\vartheta$ is only governed by the gradients in water depth:

$$C_\vartheta = \frac{\sigma}{\sinh kh}\left(\frac{\partial h}{\partial x}\sin\vartheta - \frac{\partial h}{\partial y}\cos\vartheta\right) \qquad (12)$$

## 2.3 Numerical grid

The unstructured numerical grid may consist of any combination of triangular or quadrangular cells (faces). The grid is defined by a list of grid points, with x, y and depth coordinates, and a list of cells, each with 3 or 4 node numbers. The x and y coordinates can be in a projected coordinate system (in m) or in WGS84 spherical coordinates (in decimal degrees). All values are defined in the nodes of the grid, so no staggering is applied.

This grid definition includes fully triangular meshes, rectangular or curvilinear meshes and stepwise refined rectangular meshes where the transitions to finer resolution are filled by triangles. No orthogonality is assumed. The grid can be specified as an ASCII file with node definitions and cell definitions, or as a NetCDF UGrid file, where only the node coordinates and the face connectivity are used.

## 2.4 Discretization and solution method

Equation (11) can be discretized as follows:

$$\frac{ee_{k,i\vartheta}^{n+1} - ee_{ik,i\vartheta}^{n}}{\Delta t} + \frac{c_{g,k}ee_{k,i\vartheta}^{n+1} - c_{gu,i\vartheta}ee_{u,i\vartheta}^{n+1}}{\Delta s_{k,i\vartheta}} + \frac{c_{\vartheta,k,i\vartheta}ee_{k,i\vartheta+1}^{n+1} - c_{\vartheta,k,i\vartheta-1}ee_{k,i\vartheta-1}^{n+1}}{2\Delta\vartheta} + \frac{D_k}{E_k}ee_{k,i\vartheta}^{n+1} = 0 \qquad (13)$$

where $k$ is the grid node number, $i\vartheta$ the direction bin number and $n$ the timestep/iteration number. The subscript $u$ refers to the point upwind of grid point $k$, as illustrated in Figure 1; values for $cg$ and $ee$ in this point are obtained from the two points $p$, which are upwind from point $k$ for directional bin $i\vartheta$, and with weights $w$ that depend linearly on the distance between the upwind point and the two adjacent points $p$.

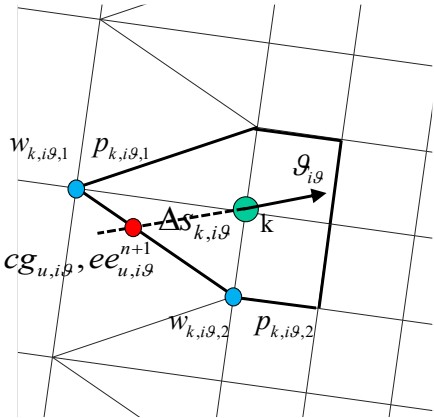

**Figure 1 Schematic showing the relation between point k and its upwind points p.**

Wave energy is conserved since we apply the conservative form of the wave energy balance. We can write the system of equations per grid point as:

$$A\,ee_{k,i\vartheta-1}^{n+1} + B\,ee_{k,i\vartheta}^{n+1} + C\,ee_{k,i\vartheta+1}^{n+1} = R(ee_{k,i\vartheta}^{n}, ee_{prev,i\vartheta}^{n+1}) \quad (14)$$

Here, the coefficients, depending on the choice for an upwind or central scheme, are given by:

*Upwind*, $c_{\vartheta,k} > 0$  *Upwind*, $c_{\vartheta,k} < 0$  *Central scheme*

$$A = \frac{-c_{\vartheta k,i\vartheta-1}}{\Delta\vartheta} \qquad A = 0 \qquad A = \frac{-c_{\vartheta k,i\vartheta-1}}{2\Delta\vartheta}$$

$$B = \frac{1}{\Delta t} + \frac{c_{gx,}}{\Delta s_{k,i\vartheta}} + \frac{c_{\vartheta,k,i\vartheta}}{\Delta\vartheta} + \frac{D_k}{E_k} \qquad B = \frac{1}{\Delta t} + \frac{c_{gx,}}{\Delta s_{k,i\vartheta}} - \frac{c_{\vartheta,k,i\vartheta}}{\Delta\vartheta} + \frac{D_k}{E_k} \qquad B = \frac{1}{\Delta t} + \frac{c_{gx,}}{\Delta s_{k,i\vartheta}} + \frac{D_k}{E_k} \qquad (15)$$

$$C = 0 \qquad C = \frac{c_{\vartheta,k,i\vartheta+1}}{\Delta\vartheta} \qquad C = \frac{c_{\vartheta,k,i\vartheta+1}}{2\Delta\vartheta}$$

In all cases the right-hand side is given by:

$$R = \frac{ee_{k,i\vartheta}^{n}}{\Delta t} + \frac{c_{gx,prev}\,ee_{prev}^{n+1}}{\Delta s_{k,i\vartheta}} \qquad (16)$$

This is a tridiagonal system with the dimension of *ntheta* that can be efficiently solved for each point using a standard Thomas algorithm. The solution for each point relies on having (ideally converged) estimates of the wave energy density *ee* in the
145 upwind points for each wave direction. For each point in the unstructured mesh, the spatial propagation is solved by backtracing, for each direction, to the line connecting two upwind points, in a manner similar to STWAVE and unSWAN. The combined propagation, refraction and dissipation are solved implicitly for each point. Wetting and drying is handled simply by making points inactive that have $\int_{0}^{2\pi} aa\,d\vartheta$, depth less than 1.1 times *hmin*, set to 0.1m by default.

To arrive at a stationary solution, we set the time step to a very large number and apply a 'sweeping' technique as follows. We
determine for each sweep s and each point *k* the distance $r_{k,s}$ along the mean wave direction $\vartheta_m$, the two orthogonal directions and the opposite direction as follows:

$$r_{k,s} = x_k \cos\left(\vartheta_m + \frac{\pi}{2}S_s\right) + y_k \sin\left(\vartheta_m + \frac{\pi}{2}S_s\right), \quad all\ k,\ s = [1:4] \quad (17)$$
$$S_s = [0, 1, -1, 2]$$

Next, we sort the points in all four directions and store the index for all points and each sweep direction. For each sweep, this
index determines the order in which we solve equation (14).

Generally, the first sweep already solves a major part of the wave propagation, as forward-marching techniques would do.

Secondary effects of refraction are covered by 'sweeping' in all 4 directions. Since the wave dissipation is a very nonlinear function of the wave height and water depth, the whole system needs to iteratively come to a converged solution. Convergence is checked after all four sweeps in an iteration; points, where the maximum difference in energy density divided by the maximum energy density for that point is less than a user defined threshold *crit*, are fixed and taken out of the loop. The iteration is converged when the maximum difference in energy density, normalized by the maximum energy density, is below the same *crit*. As the method converges rapidly and we take out the points already converged, we can set the default *crit* at a comfortably low value of $10^{-5}$. The process generally converges within 4-6 iterations.

## 3. Verification

In comparison of model to theory or data, the error metrics Pearson's correlation coefficient (rho), scatter index (sci), relative bias (relbias), and Brier skill score (skill) are computed as shown in Table A.1.

### 3.1 Linear shoaling and refraction

A first test of the correct implementation of refraction and shoaling is to compare the wave height and mean wave direction over a longshore uniform, double barred profile. We use an analytical representation of typical Dutch barred profiles (Bakker and De Vroeg, 1988):

$$z_b = z_r - Ax^b - A_b e^{-\left(\frac{x-x_b}{R_b}\right)^2} \cos\left(2\pi\left(\frac{x}{L_b} - \frac{t}{T_b}\right)\right) \quad (18)$$

With $z_b$ the bed level, $z_r$ a reference level of 6 m, $A_b$ the bar amplitude of 1 m, $R_b$ the bar scale of 200 m, $x_b$ the location of maximum bar amplitude (300 m), $L_b$ the bar wavelength (200 m) and $T_b$ the bar migration period (10 yr). The expression describes a bar system that grows, migrates seaward and damps in a periodic fashion; the time $t$ was taken arbitrarily as 0 yr, which means that the bar crest is at the location of maximum amplitude. The water level was set at 0m, and as purely refraction and shoaling, no breaking, were tested, the depth was cut off at 1 m and the breaker parameter *gamma* set to a high value..

Three grid configurations were applied: one with a uniform grid size of 20m by 20m (denoted 'uniform_20'), one with uniform resolution of 10m by 10m (denoted 'uniform_10') and one where the resolution varied from 40 m to 10 m through two uniform refinements ('variable_40_10'). The domain was 2,000 m cross-shore by 10,000 m cross-shore; the coarse uniform grid had 50,000 nodes, the fine uniform grid had 200,000 nodes and the non-uniform grid approximately. 31,000 nodes.

Uniform boundary conditions were specified on the offshore boundary and Neumann boundary conditions (no longshore gradient) at the lateral boundaries. The boundary conditions and model settings are specified in Table 1

| Parameter | Value |
|---|---|
| Hm0 significant wave height (m) | 1.0 |

| Tp peak period (s) | 5.0 |
|---|---|
| Mean wave direction (º from shore normal) | 0,30,45 |
| Directional spreading (º) | 5 |
| Directional resolution (º) | 1 |
| Directional sector (º) | 180 |

**Table 1 Parameters shoaling and refraction test**

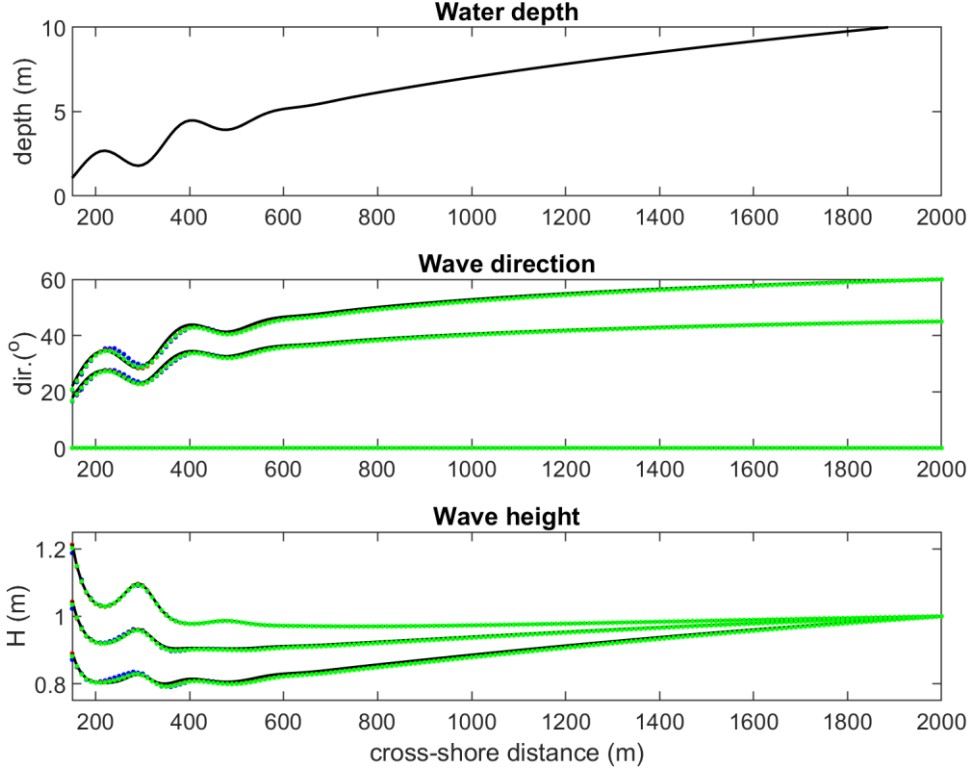

**Figure 2 Refraction and shoaling test; comparison between analytical solution for 0, 45 and 60 degrees angle of incidence (drawn black lines) and SnapWave results for uniform grid (blue dotted line), fine uniform grid( red dotted line) and unstructured grid (green dotted line).**

The results show a good agreement between the analytical and numerical wave direction and wave height, as is shown in Figure 2 and in the statistics Table A.2 and Table A.3, with a scatter index of less than 1% for the wave height and wave direction, and a bias in wave height of less than 1% and less than 1.2% in wave direction. As expected, the model that is refined in the barred area has slightly better skill than the coarse uniform grid, comparable to the fine uniform grid, at less than a sixth of the number of nodes; no deviations are found at the transitions in grid resolution.

## 3.2 Circular island

The circular island testcase is included to illustrate the capability of SnapWave to compute the wave refraction and shoaling all around an island, and to show how the solution scheme progresses. The conditions are taken from the case of a sandy circular island (Kamphuis and Nairn, 1985), with a radius of 350 m, and a 1:12 slope until a depth of 20 m. A circular curvilinear grid was applied with uniform cross-shore resolution of 5 m; the directional resolution was 5º and the sector was 360º. Wave conditions were imposed using an Hm0 wave height of 2m, a peak period of 15 seconds and directional spreading

of 20º. Various angles of incidence were tried, all uniformly applied on the outer boundary, resulting in symmetrical patterns. In Figure 3 the computational grid and bathymetry are shown, as well as the wave height distribution for incident wave angle of 270ºN. The resulting focusing of the waves in front of the island, and the reduced wave height on the sides and in the back of the island agree well with earlier results shown by (Roelvink et al., 2013) for XBeach, both in stationary and nonhydrostatic mode.

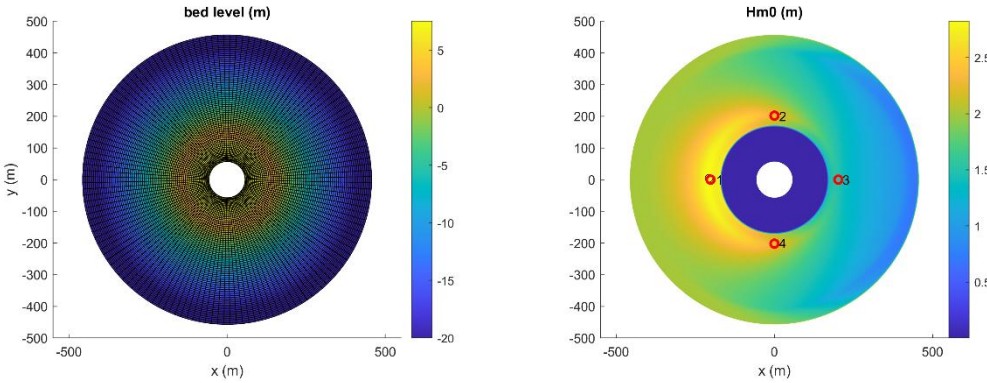

**Figure 3 Circular island test. Left panel: bathymetry and computational grid. Right panel: wave height Hm0 distribution. Points 1-4 correspond to locations of directional distributions.**

The sweeping process converges rapidly and resolves the wave pattern all around the island. In Figure 4 the directional

distributions of the directional energy density *ee* are shown for the first 5 sweeps, at four points surrounding the island. As the first sweep is plotted last, a purely green line indicates that all subsequent sweeps are hidden behind it and therefor have not changed much. This is clearly the case for point 1 on the windward side, where the first sweep going from East to West is almost fully converged. In point 4, sweep 2 going from South to North almost fully resolves the distribution. It modifies the peak in point 2 but not completely, and it adds the purple peak in point 3 at the leeward side. Sweep 3 proceeding from North

to South produces the second peak at point 3, and brings the peak in point 2 to the same level as in point 4. In point 3 at the

lee of the island, sweep 4 brings the peak at around 40 ° at the final level. Subsequent sweeps and iterations have very little impact and quickly converge to high accuracy, as indicated in Table 2.

**Table 2 Convergence characteristics of circular island test.**

| Iteration | Maximum error (%) | Percentage of fully converged points (%) |
|---|---|---|
| 1 | 1.00000 | 29.63 |
| 2 | 0.03473 | 34.24 |
| 3 | 0.00009 | 97.97 |
| 4 | 0.00000 | 100.00 |

An interesting aspect of the solution is that at the leeward side we have waves from almost opposing directions. In the nonhydrostatic solution in (Roelvink et al., 2013) this was also observed.

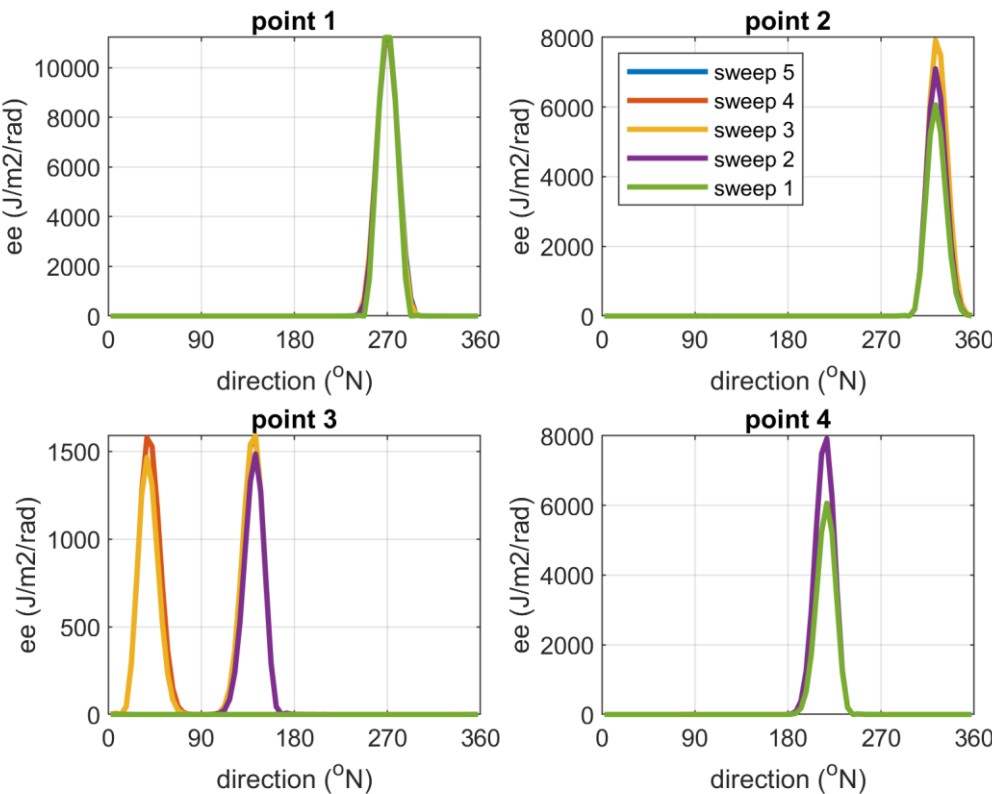

**Figure 4 Circular island test. Change in directional distribution after first 5 sweeps, first one plotted last, for the four indicated**
**points.**

We can conclude that the wave patterns are realistic and that the method quickly converges for waves incident from any direction.

### 3.3 Circular reef

The *circular reef* case was inspired by the work of (Mandlier and Kench, 2012) who considered analytical solutions to the refraction problem using ray tracing. The case we present has a flat circular reef with a radius of 350 m, a depth of 1.5 m on the reef and deep water (taken as 100 m) all around it.

To be able to compare our model with the analytical solution in terms of wave height distribution, we reproduced the wave ray refraction pattern as described in (Mandlier and Kench, 2012) and added the computation of wave heights, by counting the

number of wave rays passing within a certain distance, taken as 4.5 times the initial ray distance, from each grid point in a regular 5m by 5m grid, and computing the wave height as the rms value of the wave heights associated with each refracted wave ray within this distance. The incident wave height was 0.1 m, ensuring wave breaking did not play a role. The resulting refraction pattern and wave height distribution are shown in Figure 5, showing a highly concentrated wave height region around 90m East of the centre of the reef, and two areas of very low or undetermined wave heights where the wave rays cannot

reach.

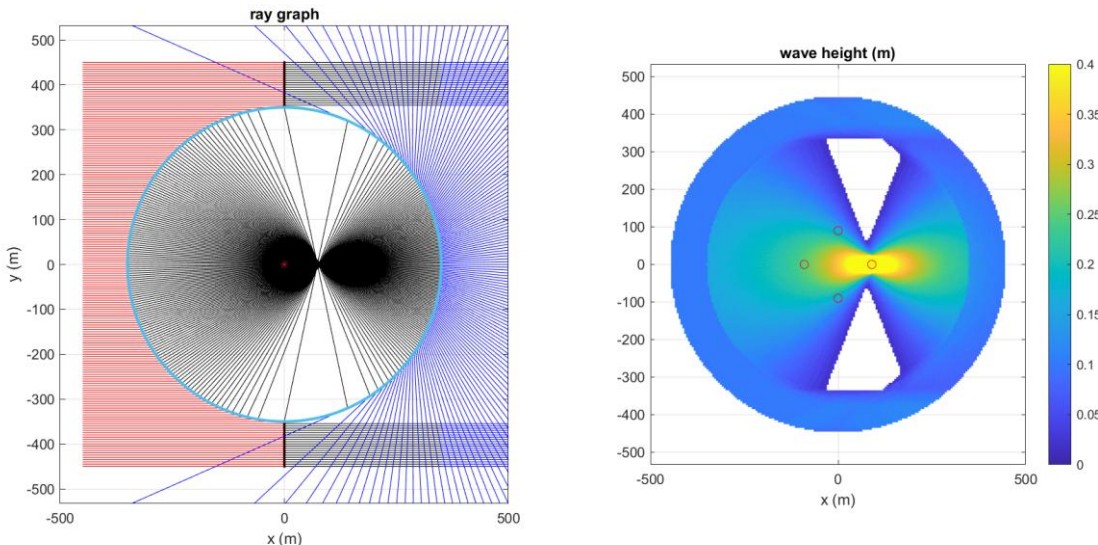

**Figure 5 Reproduction of Mandlier and Kench analytical solution for a flat circular reef; left panel: refraction pattern of wave rays; right panel: derived wave height.**

For SnapWave, we constructed a circular grid with 5m radial resolution and 1° angular resolution, except for the part within

50m of the centre, which was filled in with triangles with sides around 2.5m, resulting in the grid shown in the left panel of Figure 6. The grid was rotated to check whether the implementation was sensitive to the grid orientation, which it was not.

The wave conditions were a mean direction of 270 ºN, a peak period of 12 s and a small directional spreading of 5 º. The wave angle resolution was 1º. A small incident wave height of 0.1 m was applied to enable a comparison with the analytical solution of Mandlier and Kench (2012). In the right-hand panel the wave height distribution is shown, where we see a narrow area of concentrated wave height at around 60-150m from the centre of the reef. This corresponds reasonably well with the area of concentrated wave energy in the analytical solution, which centred around 90 m from the centre. It must be noted that our model provides seemingly reasonable results on the leeward side of this caustic and does not blow up; for higher incident waves the wave breaking mechanism kicks in and limits the growth of the wave height near the caustic.

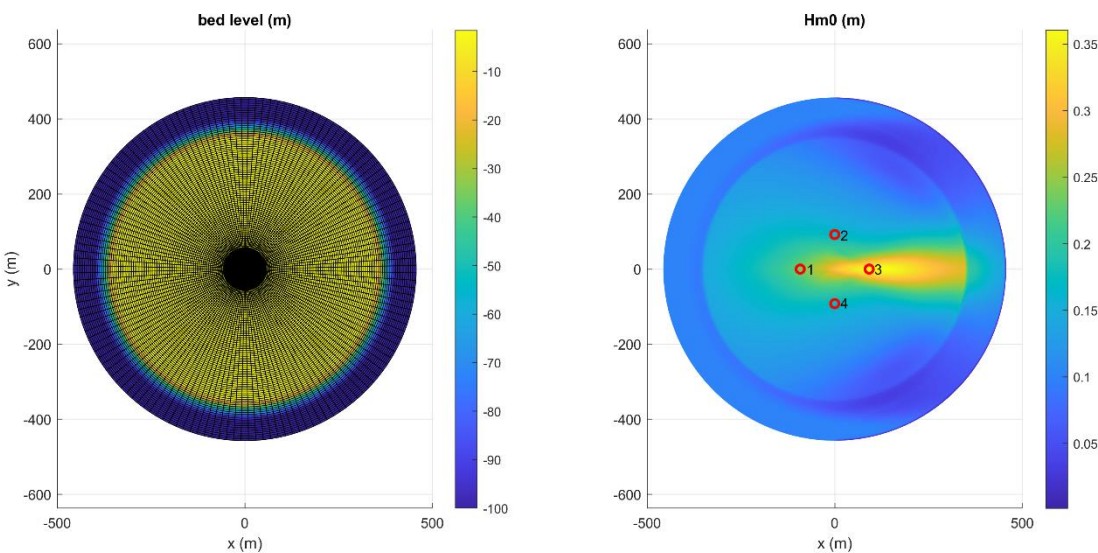

**Figure 6 Reef refraction test. Left panel: bathymetry and computational grid. Right panel: wave height Hm0 distribution. Black circle indicates maximum wave height; points 1-4 correspond to locations of directional distributions.**

The building up of the solution is almost complete in the first sweep, for points 1, 2 and 3. For points 2 and 4 the fourth, East to West sweep brings some additional energy peak from almost easterly direction, due to refractive trapping along the edge of the reef. In any case it is symmetrical and relatively small.

We may conclude that, although there is not a perfect match, the SnapWave model produces a very similar wave height pattern at the windward side and an area of highly focused wave height over an area similar to the analytical solution. Interestingly enough, the SnapWave method is considerably faster than the analytical solution.

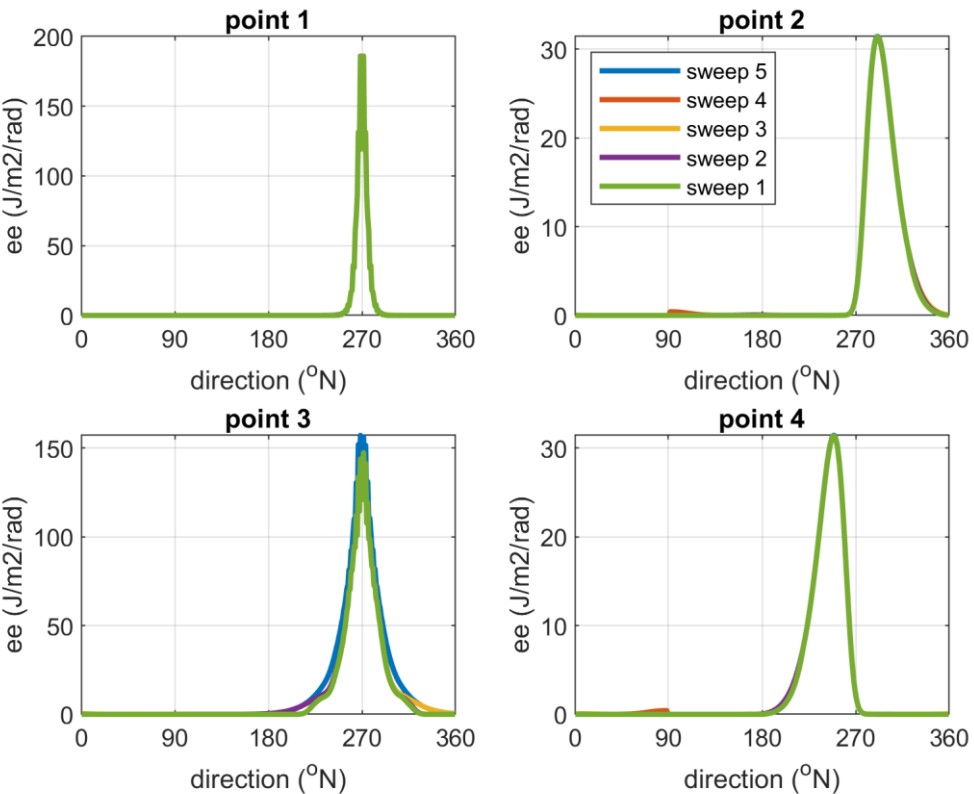

**Figure 7 Change in directional distribution after first 5 sweeps, first one plotted last, for the four indicated points.**

## 4. Field validation

The main objective of the field validation cases is to demonstrate a methodology to hindcast or predict nearshore wave conditions based on ERA5 data at locations ~0.5° offshore, global or local bathymetry and SnapWave to transform the wave conditions to specified nearshore points. We consider four testcases, spanning a range of conditions and geographical locations.

- Coast3D campaign at Egmond, the Netherlands, situated on an open, barred coast
- Ameland inlet, the Netherlands, under the influence of a large ebb tidal delta
- St Croix, US Virgin Islands, with operational buoys on either side of the island
- Ningaloo Reef, Australia, with an array of pressure sensors across a wide, shallow reef.

For all cases we use a similar setup starting from ERA5 model output points at 0.5º resolution. For two of the cases, Coast3D and Ningaloo, we compare these results with those of a local model driven by locally measured wave conditions, in order to distinguish between errors in SnapWave and those inherent in the ERA5 model.

## 4.1 Coast3D

**Local model vs dcsm-fine**

This testcase concerns the hindcasting of wave conditions at the 15m depth contour and the subsequent propagation and dissipation of the waves throughout the surf zone at Egmond, the Netherlands. These wave measurements were part of a large EU project COAST3D (Soulsby, 2001); Egmond was one of the test sites and the main campaign at this location, in November 1998, is described extensively in (Ruessink, 1999); the wave measurements are also detailed in (Reyns et al., 2023).

In Figure 8 the extent of the large-scale model is shown along with the ERA5 output points used as boundary locations. In
Figure 9 the details of both the large-scale model and the local model are shown in the area of the field campaign. The large-scale model has a resolution ranging from approximately 800m to 100m near the entire coast, with three subsequent local refinements to approximately 14m in the measurement area. It must be noted that for providing boundary conditions to coastline models typically a grid size of 100m would be sufficient, but the finer resolution is needed to resolve the breaker bars in the surf zone. The local model has a curvilinear setup with cross-shore resolution from 70m to 13 m and a longshore
resolution from 125m to 25m; in other words, the resolution in the measurement area is similar.

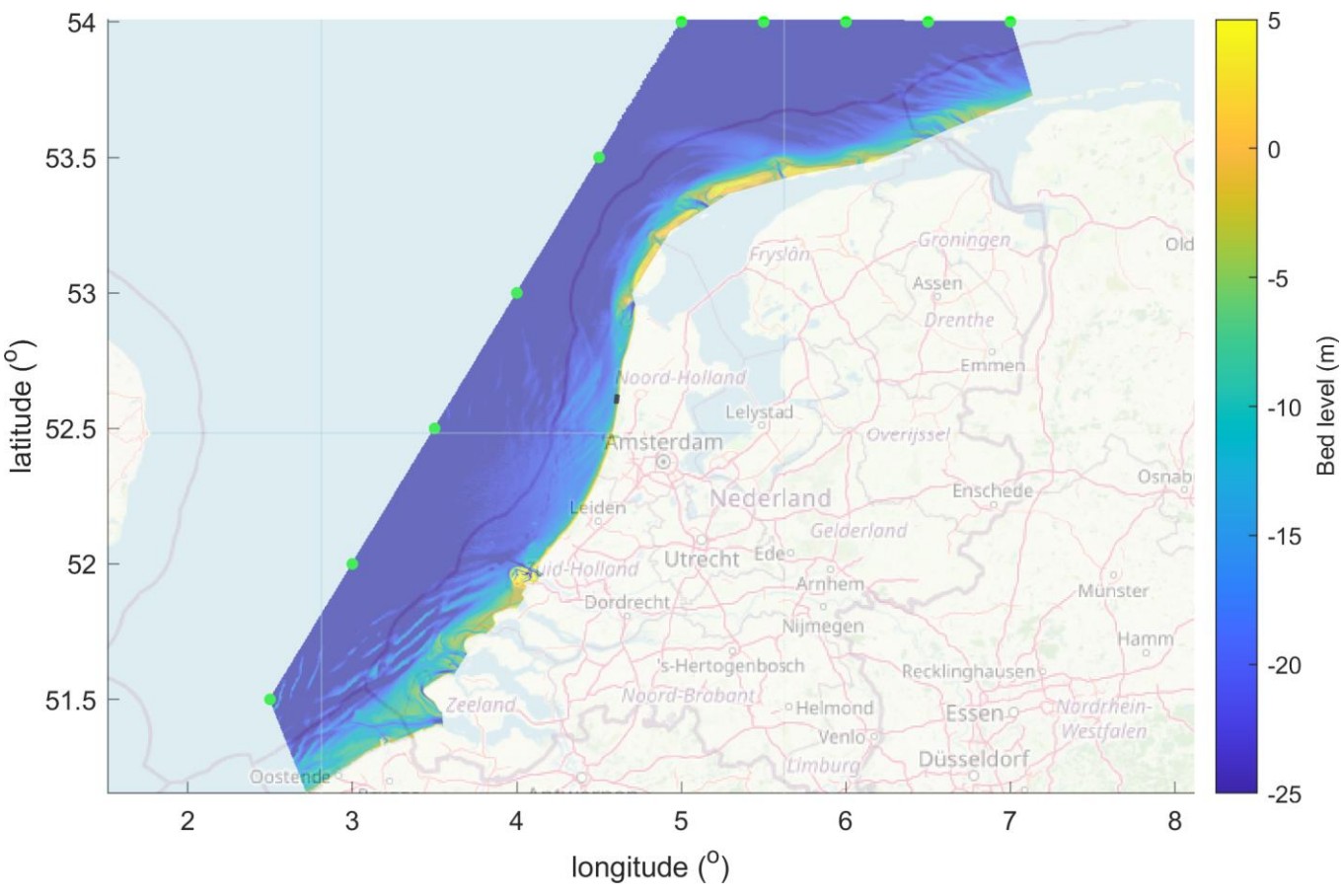

**Figure 8 Overview of Holland coast, with bathymetry of large-scale model domain; black rectangle in North Holland: location of Egmond field campaign. Green dots: locations of ERA5 boundary points.**

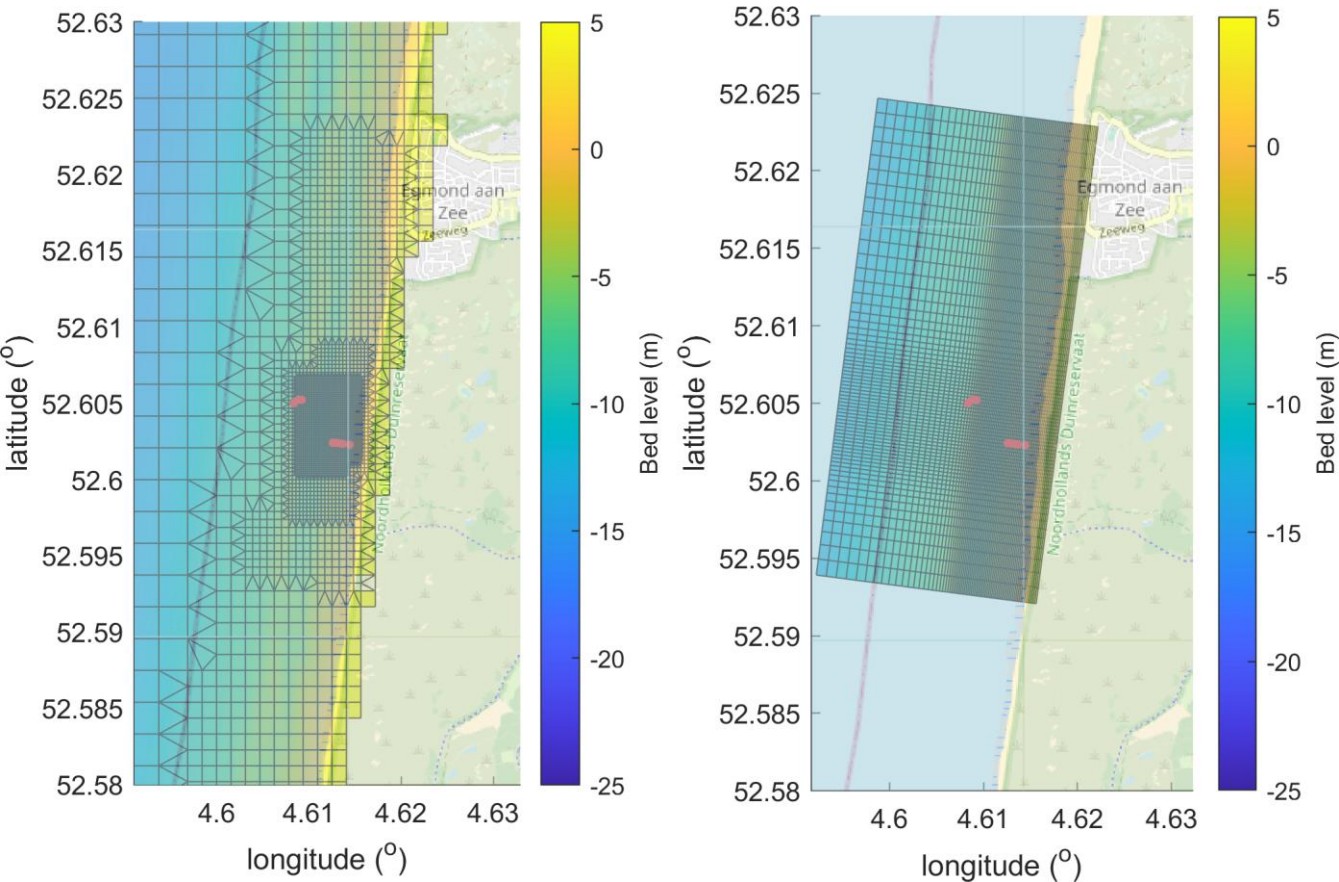

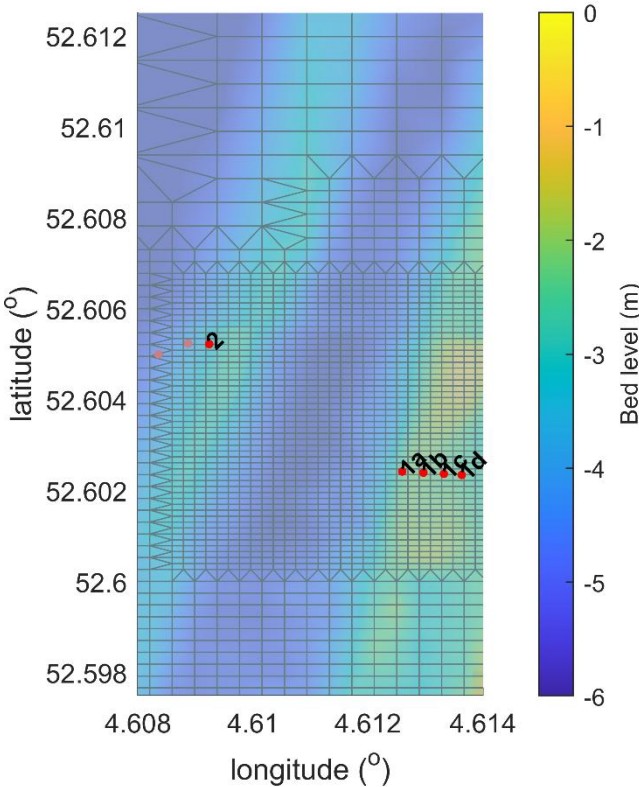

**Figure 9 Details of the computational grids for the large-scale model (upper left panel) and the local model (upper right paneland measurement locations (bottom panel).**

The measurement period considered here covered the period of Nov 1, 0:00 until Nov 12, 12:00 1998. ERA5 data were downloaded and time series were extracted for the indicated boundary points (Figure 8) at hourly intervals. Data for point 8,

at 15m depth, were used as validation data for the large-scale model, and as boundary conditions for the local model. Points 1a-1d covered the transformation over the barred profile, as indicated in (Ruessink, 1999). For the water levels, the measured timeseries, interpolated from 2 tidal stations as in (Ruessink et al., 2001) was applied uniformly over both models.

Sensitivity tests indicated that the results were little sensitive to the directional resolution, so a directional step of 10º was applied. Breaker parameter *gamma* values of 0.75 (default) and 0.70 were applied.

**Results local model**

First, we discuss the results for the local model, see Figure 10. At the outermost station 2 we see modest events on November 2,3 and 11, and a major event on November 6. At this location, the wave height variation is mostly due to the variation in offshore wave conditions. Unfortunately, only a short period is available in the observations. As we move through the surf

zone in points 1a thru 1d the effect of depth-induced breaking becomes more obvious, leading to a strong tidal modulation in the wave height time series. These results are very similar to those of (Ruessink et al., 2001) applying a profile model.

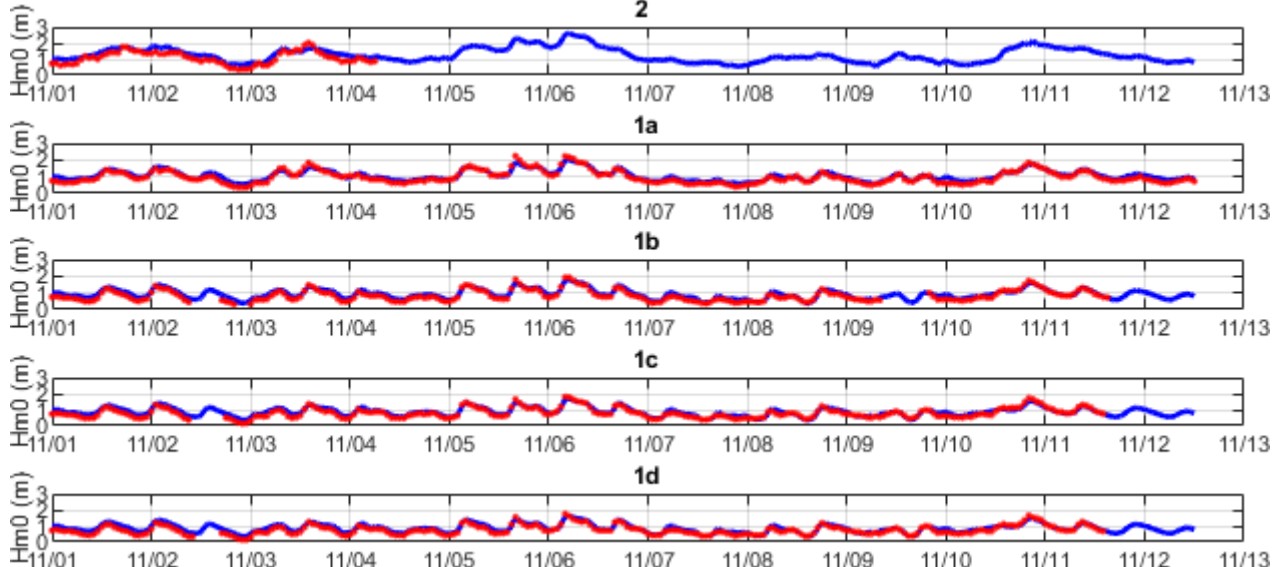

**Figure 10 Time series of Hm0 wave height at 30 min intervals, measured (red dots) vs computed (drawn blue line). Local model driven by observed wave conditions at 15m depth, *gamma*=0.7.**

In Figure 11 we show scatterplots of the computed vs. observed wave heights, for a *gamma* value of 0.7, which showed slightly less bias and higher skill than the default value of 0.75. Error metrics for this test are given in Table A.4. The results for station 2 show the highest bias and scatter, but it must be noted that these points only cover a short period. The surf zone points 1a thru 1d show a modest scatter index in the order of 15% and a bias of around 10%. Reducing *gamma* further reduces the bias but results in poorer performance for the higher wave conditions. Note that we did not take the substantial bed level changes in the inner surf zone over the course of the measurement campaign into account. Propagating the wave heights through the surf zone is performed with a skill of over 96%.

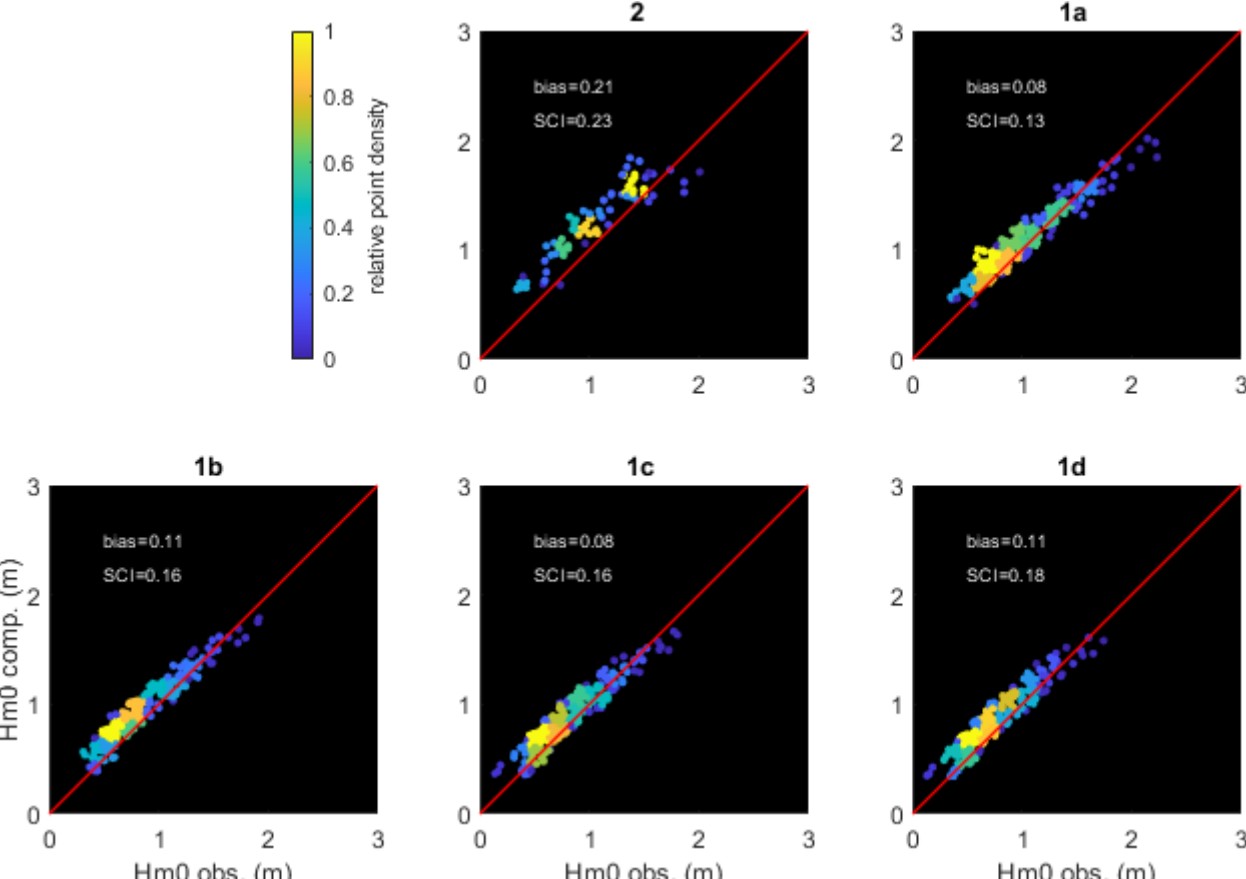

**Figure 11 Scatterplots (heat maps) of computed vs observed Hm0. Local model driven by observed wave conditions at 15m depth,** *gamma***=0.7.**

### Results large-scale model

The time series of the large-scale model simulation are shown in Figure 12. First, as an indication of the quality of the ERA5 hindcast, the observations at the point 8 at 15m water depth are generally reproduced quite well, except for a small event at November 3$^{rd}$, which is completely missed by ERA5; during that period, both wave directions and wind directions are offshore in ERA5 so there is no possibility of getting such nearshore wave heights in the order of 2 m. The other peaks are generally predicted well, sometimes with a phase shift in the order of a few hours.

The results through the surf zone, though less accurate than for the local model, generally reproduce the observed time series quite well, particularly around the main event between November 5 and 7.

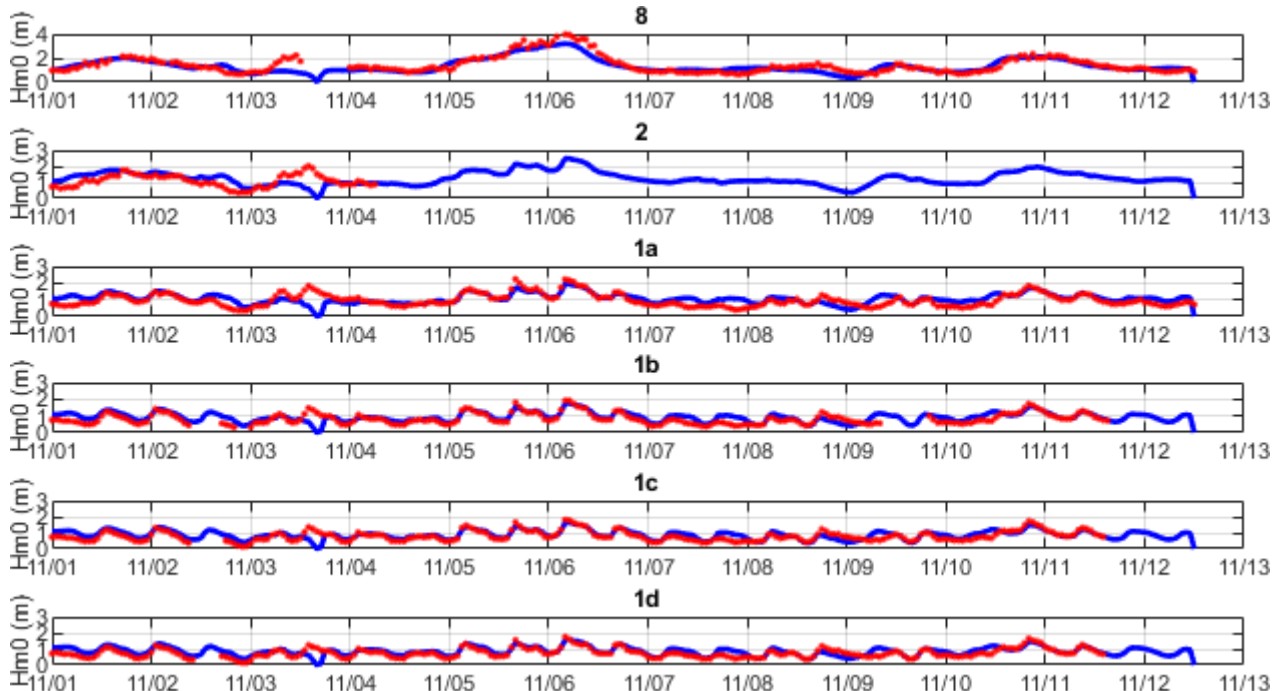

**Figure 12 Time series of Hm0 wave height at 30 min intervals, measured (red dots) vs computed (drawn blue line). Large-scale model driven by ERA5 boundary conditions, *gamma*=0.7.**

The scatterplots shown in Figure 13 confirm this narrative, as do the error metrics in Table A.5 and Table A.6. Point 8 is mostly indicative of the skill of the ERA5 hindcast and has a low bias of 7% and a scatter index of 22%. In point 2 the scatter index is quite high since the short time series includes the event that was missed by ERA5. The points through the surf zone have a

345 bias of around 10% and higher scatter indices than the local model, mostly for missing the event on November 3[rd].

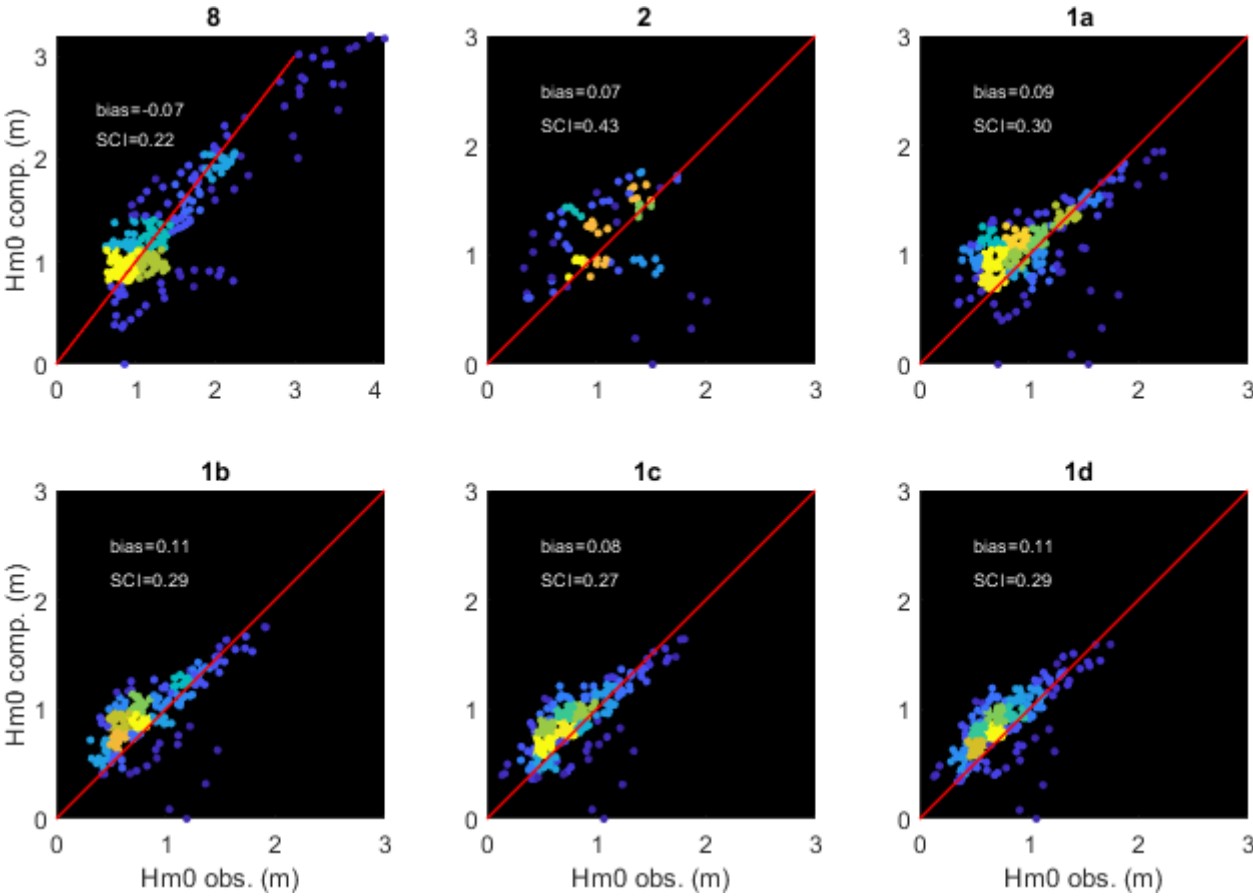

**Figure 13 Scatterplots (heat maps) of computed vs observed Hm0. Large-scale model driven by ERA5 boundary conditions, *gamma*=0.7. For color legend refer to Figure 11.**

## Conclusions

From this case study we may conclude that SnapWave has an adequate skill of more than 95% in propagating waves through the surf zone; for given wave conditions at 15m depth, the bias is in the order of 10% and the scatter index in the order of 15%. When using it to propagate waves from the nearest ERA5 output points, the bias at the 15m depth point was a low 4% and the scatter index 21%. For the subsequent propagation through the surf zone the bias remains low at around 10% and the scatter index is less than 30%; the higher values for the scatter index compared to the locally driven model are mostly due to the phase shifts between observations and model at the 15m depth point, and due to one event being missed by ERA5. According to the used definition, the model skill is around 0.9 and higher.

## 4.2 Ameland

Ameland Inlet is typical for tidal basins with a large tidal inlet, and the wave refraction, shoaling and dissipation over such ebb delta is important for the coastline evolution along the seaward-facing coast. In the framework of the project SBW (Strength and Loads on Coastal Defences) a number of directional wave riders was installed from 2007 onwards; for an overview see (Elias, 2017). Several studies have focused on the wave penetration into the Wadden Sea and on the effect of wave growth by wind and current refraction, but for the purpose of this study we focus on the wave distribution around the ebb delta. We extracted two-month time series from the MATROOS system used by the Dutch government and knowledge institutes, for the period of November 1 thru December 31, 2008. As there was uncertainty over the location of one of the buoys in early November we used the period of November 5 until December 31, 2008.

We created an unstructured grid covering all the Dutch Wadden islands and extending to the nearest reliable (i.e. not affected by land) ERA5 points, as indicated in Figure 14. The resolution ranged from 800m offshore to approximately 100m in the nearshore. The bathymetry in the area of interest was updated with area soundings ("Vaklodingen") from 2008. Six observation points were selected, as shown in Figure 15. The two points AZB11 and AZB12 are outside the ebb delta and are indicators of the quality of the ERA5 hindcast. The other four are spread out over the ebb delta and should give an impression of the quality of the wave propagation model in a complex area with shoaling, refraction and wave breaking.

The tide level was imposed uniformly based on a nearby output location from the Global Tide and Surge Model (GTSM, (Muis et al., 2016).

From ERA5 the data for Hm0 wave height, peak period, mean wave direction and directional spreading were extracted. The data for the observation points contained Hm0 wave height and Tm10 wave period, which based on our experience we converted to peak period by multiplying by 1.1.

Default parameter settings were chosen with *gamma* of 0.75, a directional resolution of 10º and a directional sector of 360º.

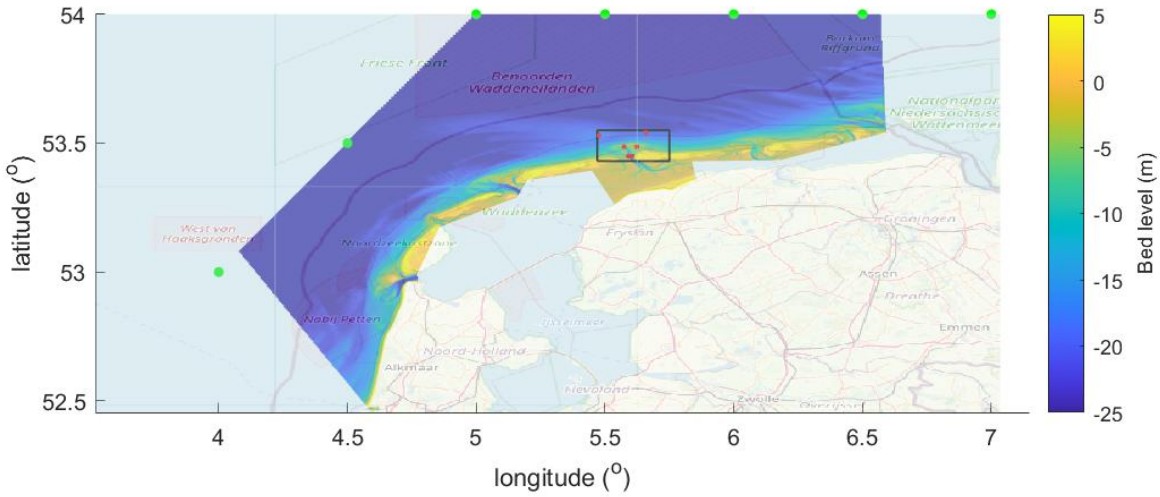

.

**Figure 14 Overview of model domain and bathymetry for Ameland hindcast. Green dots indicate ERA5 boundary points; red dots observation points.**

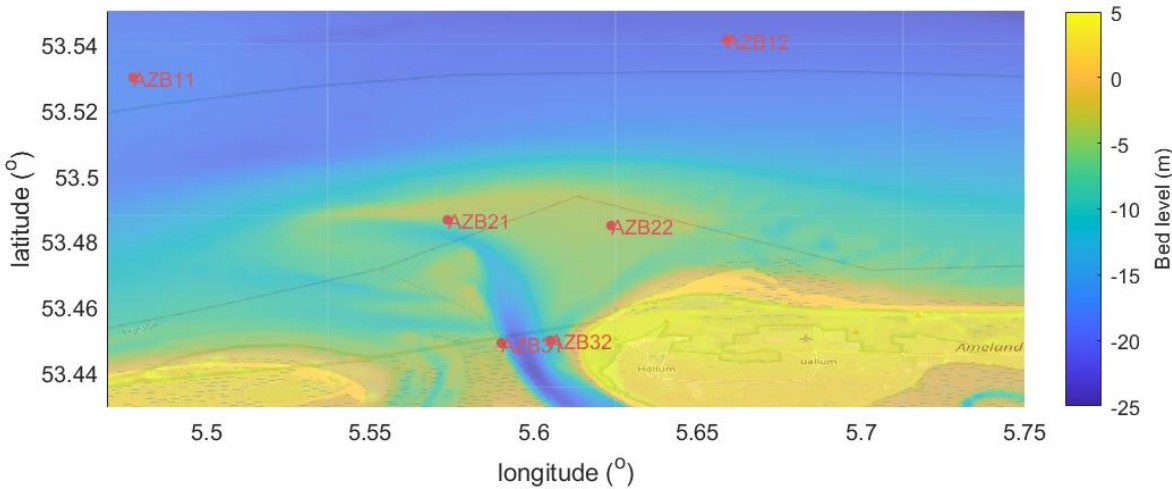

**Figure 15 Detail of bathymetry and observation points Ameland inlet.**

Time series for Hm0 wave height are shown in Figure 16. Clearly, the points AZB11 and AZB12 are in depths where the tidal modulation does not play a role yet, and generally the computed wave heights follow the measurements closely, except for the event around 22/11 where the wave height is clearly underestimated. This is likely due to an underestimation of wave heights by ERA5 for this event, though some additional wave growth due to wind (not included in this SnapWave model) could play a role as well.

The results for the four other points are clearly modulated by the tide, as wave breaking plays an important role. A change of *gamma* value to 0.8 improved the error statistics somewhat. Relative bias and scatter index are in the same order of magnitude as for the Coast3D case, around -10% and 25-30% respectively.

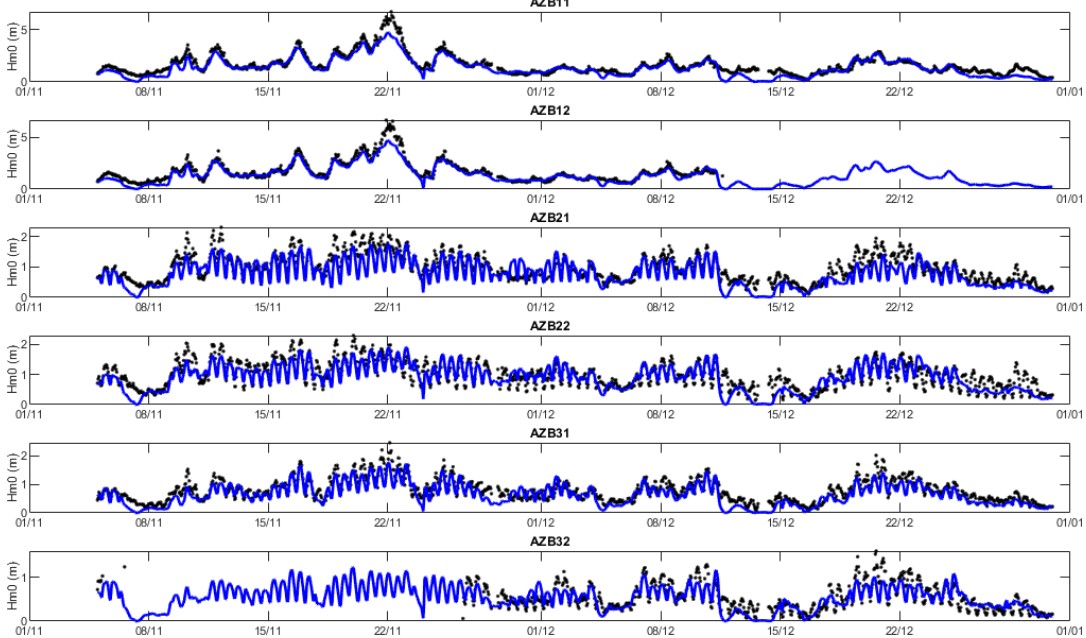

**Figure 16 Time series of Hm0 wave height for 6 observation points, Ameland Inlet.**

Though SnapWave without wind growth terms assumes a uniform distribution of the peak period, it is useful to test this assumption against the wave data. Figure 17 shows that this is not a very bad assumption; from the error statistics in Table A.7 and Table A.8 we see that the relative bias is around zero and the scatter index in the order of 25%.

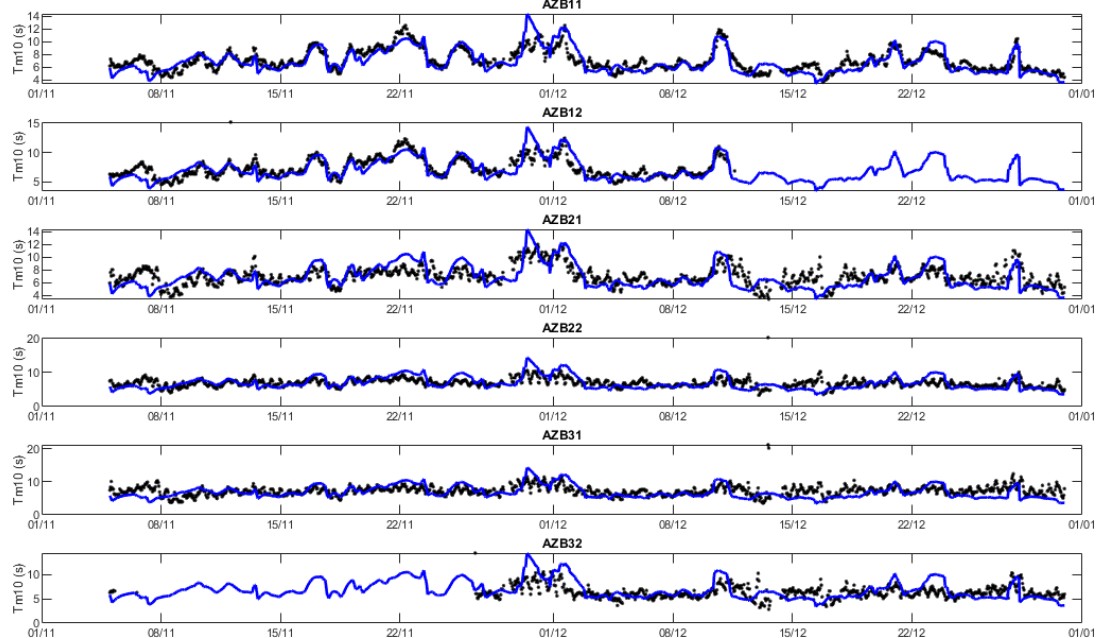

**Figure 17 Uniform Tp vs observed 1.2\*Tm01, Ameland Inlet.**

The scatterplots in Figure 18 confirm that the systematic underestimation of the higher wave heights originates with the ERA5 data and propagates through the nearshore area. The scatter in the nearshore points is rather consistent at around 30%. According the used definition, the model skill is consistently over 0.9, indicating an adequate performance for such cases.

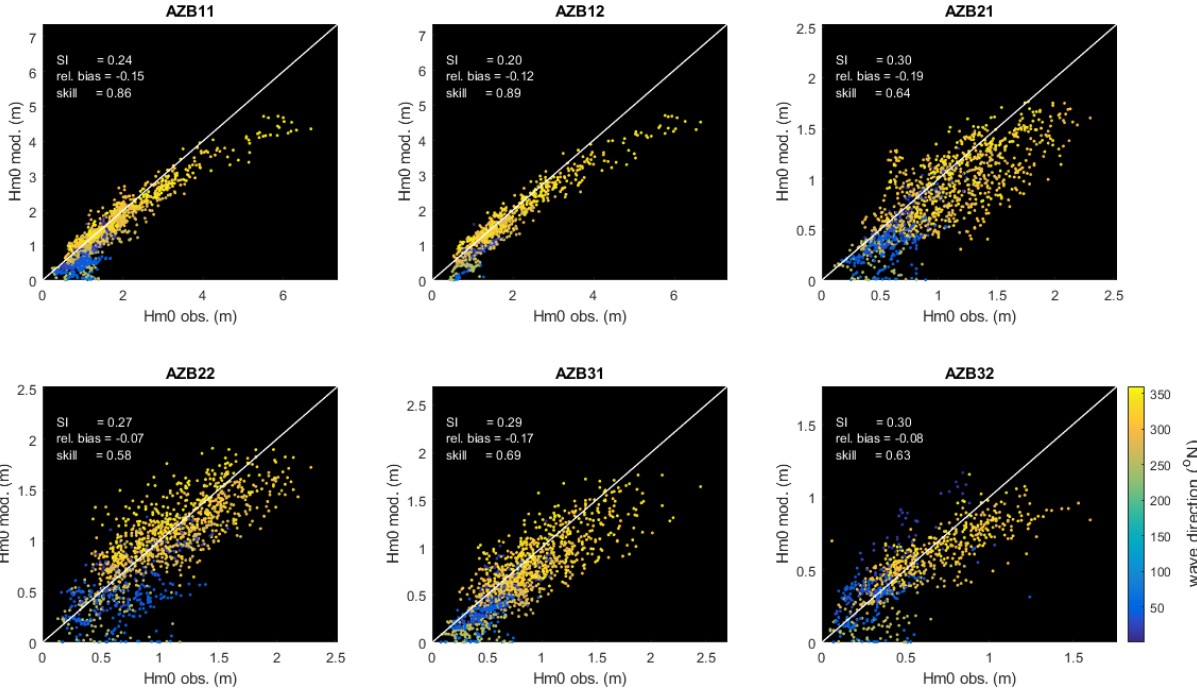

**Figure 18 Scatterplots SnapWave Hm0 vs observations, coloured by wave direction.**

The performance for this model is similar to that of the Coast3d large-scale model; the number of nodes is around 250,000 and the run time per wave condition is around 2.5s, or approximately 10 microseconds per node and wave condition.

## 4.3 St Croix

The island St Croix (US Virgin Islands) is used as a case study where open boundaries are applied at all sides, and ERA5 data are specified all along these boundaries. There are two operational CDIP buoys (https://cdip.ucsd.edu/m/about/ ) at the edge of the shelf, called 'Fareham' on the southern end and 'Christiansted' on the northern side. In terms of processes needed, the case is not too challenging. We test mainly if the ERA5 hindcast is accurate and if the shielding and for some wave directions the refraction on the shallow reef areas are properly accounted for. The model setup is shown in Figure 19; the cell sizes range from 800m offshore to 200m near the coast; higher resolution was not needed here as the observation points were on the edge of the shelf, still in relatively deep water. We obtained wave height and period records from the CDIP buoys for the period of June 1 until November 1, 2010, and downloaded ERA5 wave data for Hm0 wave height, peak period, mean wave direction and directional spreading for the same period, for the locations indicated by the green dots.

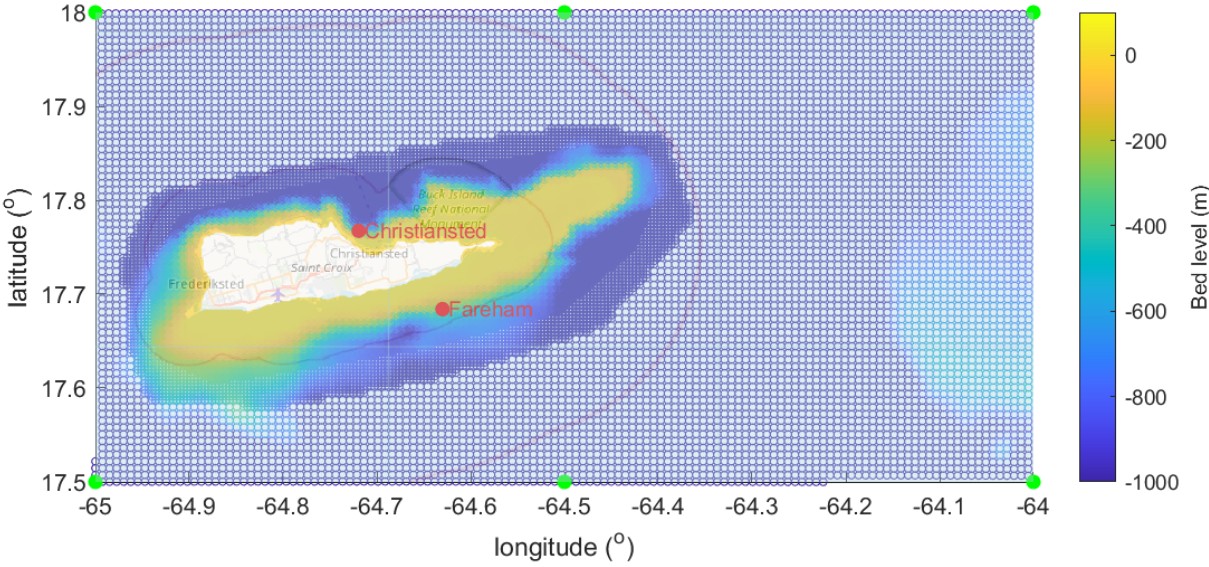

**Figure 19 Grid layout and bathymetry, St Croix case. Green dots indicate ERA5 boundary locations; red dots indicate the observation points at the site of the CDIP buoys.**

In Figure 20 the time series comparison is shown for the two observation points. In general, the model follows the observations closely, except for the event on September 18th which is severely underestimated at the Christiansted buoy, and an event on

October 6th, which the model underestimates at the Fareham buoy. Such behaviour where ERA5 misses some of the extreme

peaks due to a lack of resolution is well documented (e.g. (Fanti et al., 2023)).

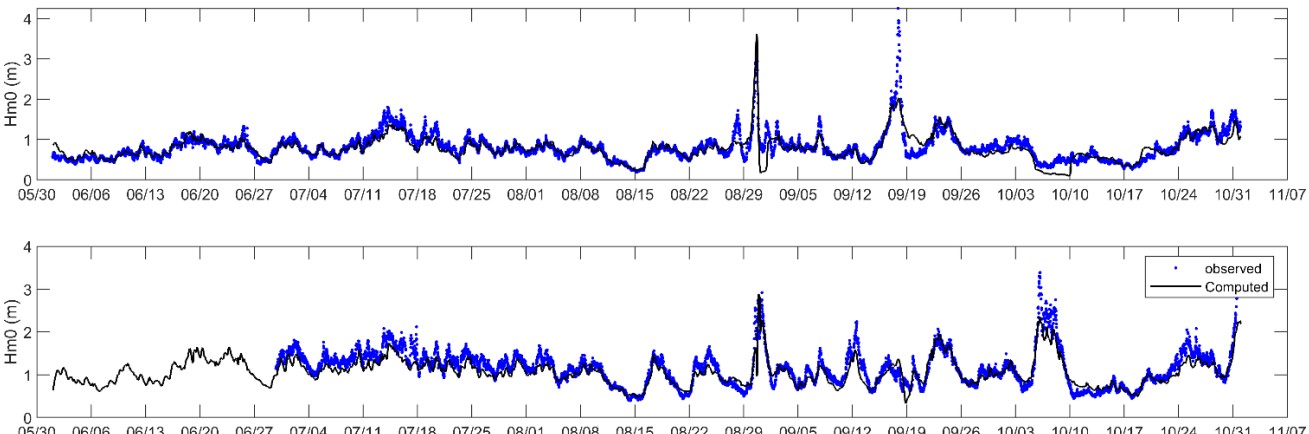

**Figure 20 Time series of Hm0 wave height for stations Christiansted and Fareham near the island of StCroix, computed (drawn black line) vs. observed (blue dots).**

The scatterplots (heat maps) confirm the fact that for most conditions the agreement is quite good, and only for some individual

events the ERA5 model misses the peaks. Overall, as is also apparent from the error statistics, the bias is less than 10% and

the scatter index in the order of 20%, and skill over 95%.

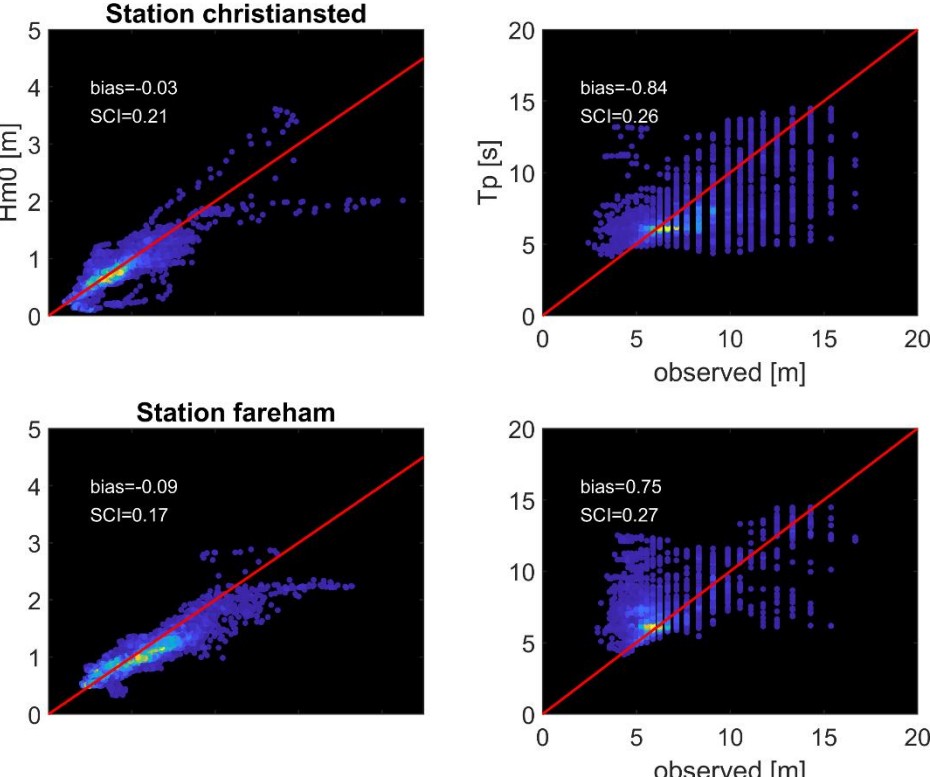

**Figure 21 Scatter plots (heat maps) of computed vs. observed Hm0 wave heights (left panels) and Tp wave period (right panels), for stations Christiansted and Fareham, St Croix. For color legend refer to Figure 11.**

The computation for the 5 months took 51 minutes, on average 0.8 s per wave condition or 20 microseconds per node per wave condition (TBD: clean performance check)

## 4.4 Ningaloo Reef

Ningaloo Reef is a wide and extensive, pristine coral reef in NW Australia. The reef has been the subject of a number of studies on hydrodynamics and sediment transport, and data collected there (Pomeroy et al., 2012) has been used to validate other wave models such as XBeach in (Van Dongeren et al., 2013). That study focused on the generation of infragravity waves but also considered the propagation of the swell waves as we do here. One important finding in these studies was that the roughness of the reef was very high, and could be mimicked by using a high friction factor $fw$ of 0.6 on the reef. Here we used this value, making use of the option to impose space-varying roughness fields as random samples. We focus in a cross-shore transect with pressure sensors C1 on the forereef and C3 through C6 on the reef flat.

We used two model setups: one local model (square cells, resolution 16m by 16m) driven entirely by locally measured wave conditions, and one unstructured grid with square cells, refined 5 times, with resolution from 500m to 16m. The overall model grid is shown in Figure 22 and details at the measurement site are shown in Figure 23. For the water level in the large-scale model we extracted time series for a nearby location from, the GTSM (Muis et al., 2016) for the month of June 2009  and imposed this uniformly.

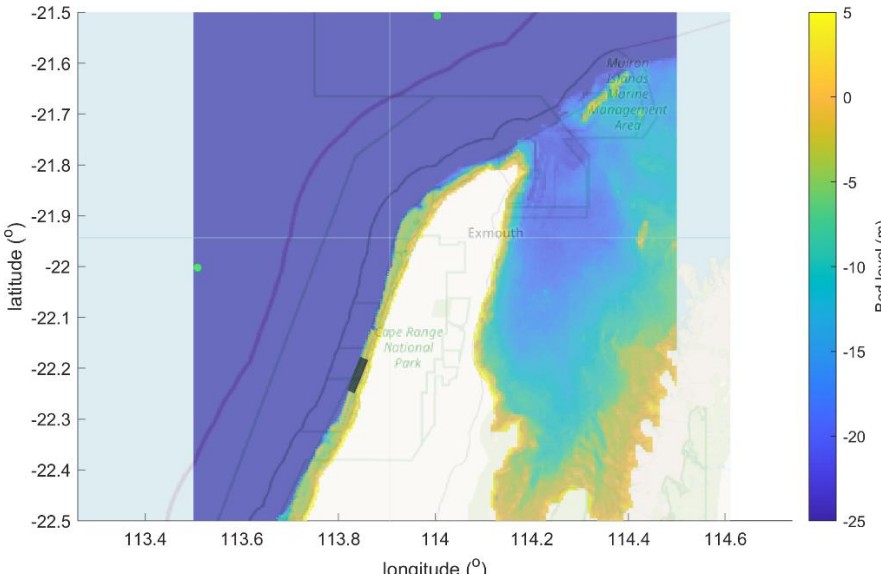

**Figure 22 Overview of large-scale Ningaloo reef model with bathymetry, boundary points (in green dots) and the location of the local model.**

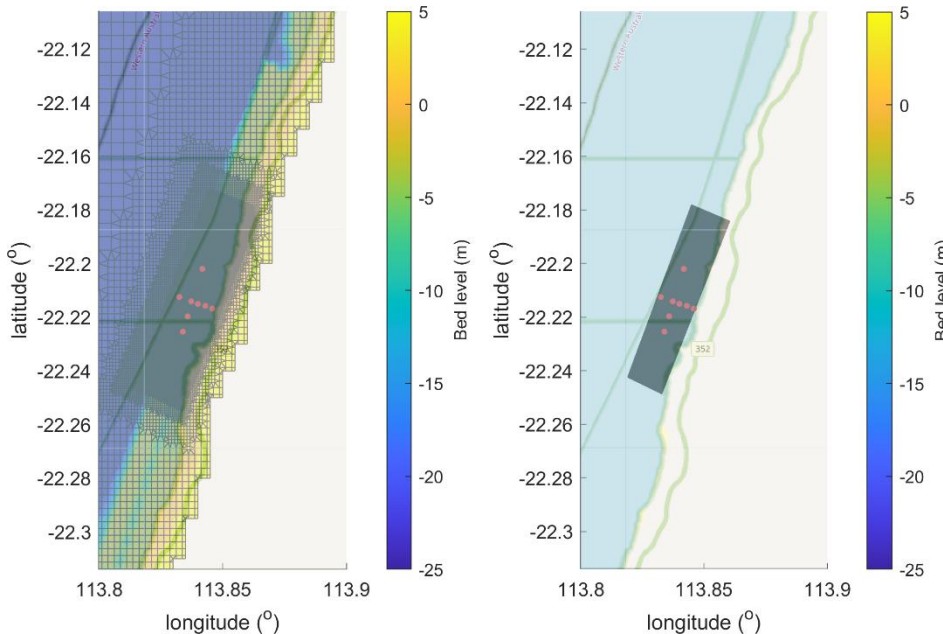

**Figure 23 Detail of large-scale model at measurement site (left panel) and local model (right panel). Observation points in rod dots.**

First, we compare time series of the Hm0 wave height for the local model. As noted in the literature, the swell heights rapidly decay behind the reef edge, a process that is dominated by the bed friction. Also, as in the Coast3D and Ameland cases, the wave heights over the reef flat are strongly modulated by the tidal water level elevation. The model results follow the observations reasonably well, given that the wave heights decrease by an order of magnitude. As SnapWave by itself does not consider the wave setup, water depths on the reef flat are underestimated, which is apparent particularly in the most shoreward points. As shown in the statistics, the bias is in the order of centimetres; the relative bias is in the range of 0-25% as the mean Hm0 on the reef flat is very low. The same holds for the rms error, which is a few centimetres, whereas the scatter index is in the order of 30%.

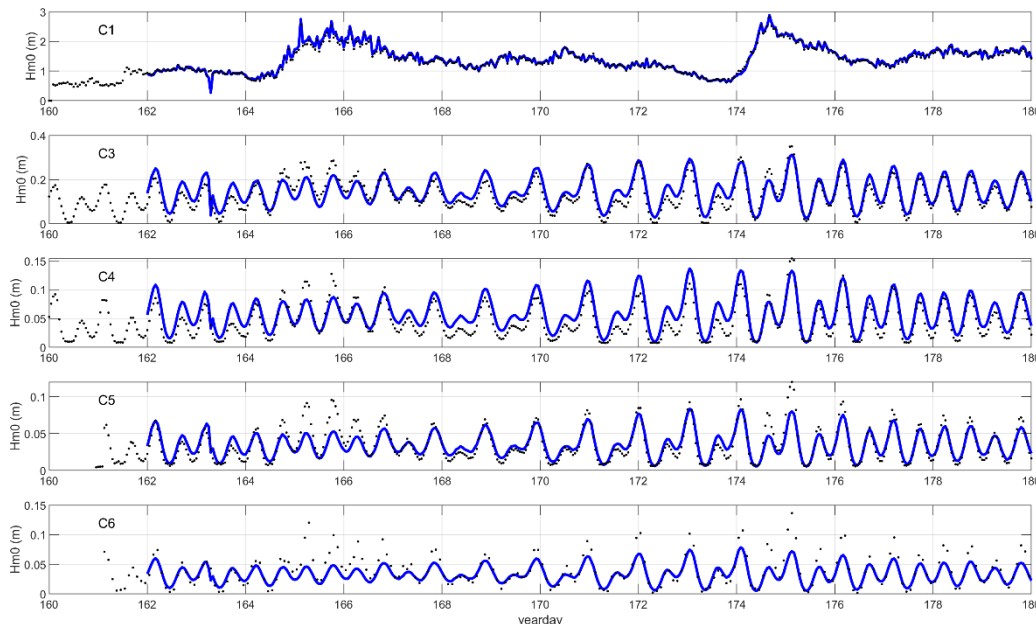

**Figure 24 Time series of HM0 wave height across the reef at Ningaloo; observations (black dots) against model simulation (blue drawn lines). Local model.**

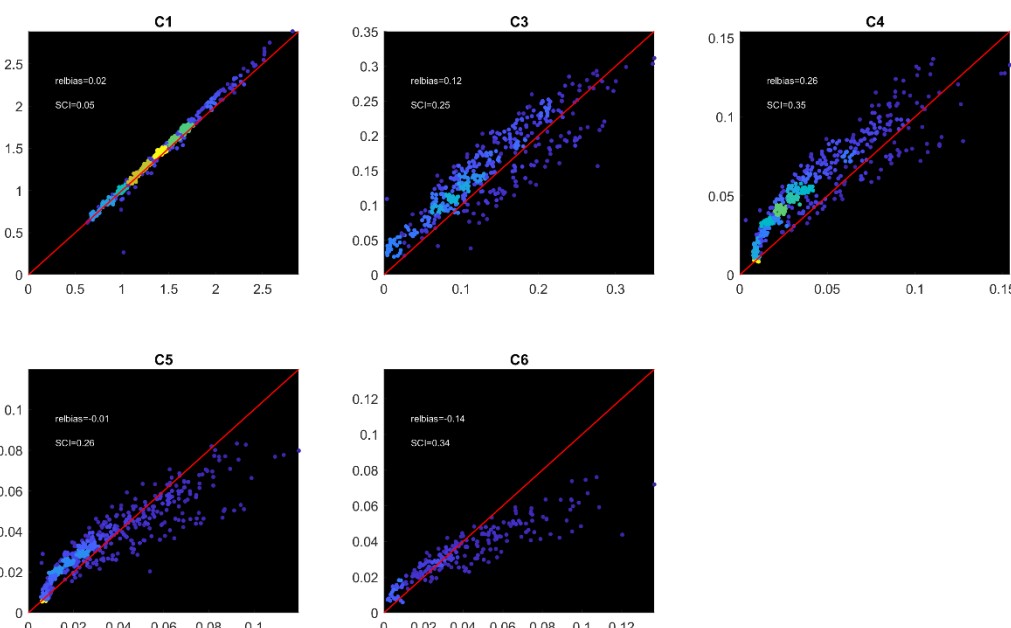

**Figure 25 Scatterplots (heat maps) of computed vs. observed Hm0 wave heights. Local model. For color legend refer to Figure 11.**

For the large-scale model the results are shown in Figure 26 and Figure 27. First, we see that the ERA5 model predicts the general trend in the wave height time series at the outer reef location, but underestimates the Hm0 around yearday 164, and overestimates for much of the remainder of the period, particularly around yearday 174. Still, the relative bias of 10% at this location and the scatter index of 29% are in line with the other case studies.

For the reef locations the relative bias is less than 10% for most locations except C4, but the scatter index is rather high, at 40-
500 65%. This is mostly due to a small phase shift between the GTSM hindcast water level and the observed water level in situ, as can be seen in Figure 28. When we apply this shift to the simulated model results and compare them with the observations, as shown in Figure 29 and Figure 30, the skill over the reef flat improves considerably, from 0.72 to 0.84.

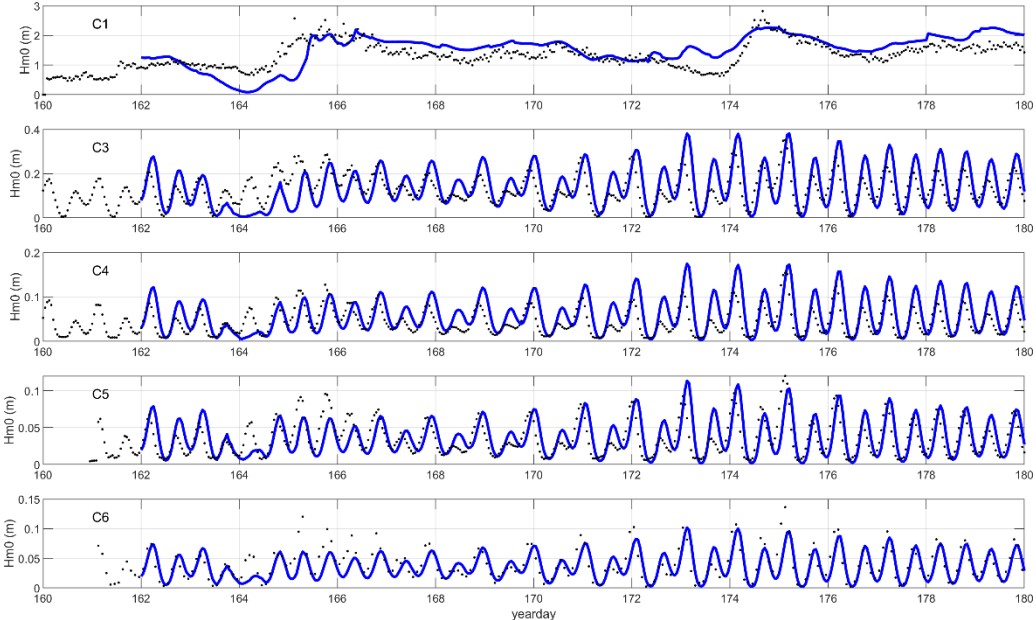

**Figure 26 Time series of HM0 wave height across the reef at Ningaloo; observations (black dots) against model simulation (blue**
**drawn lines). Large-scale model.**

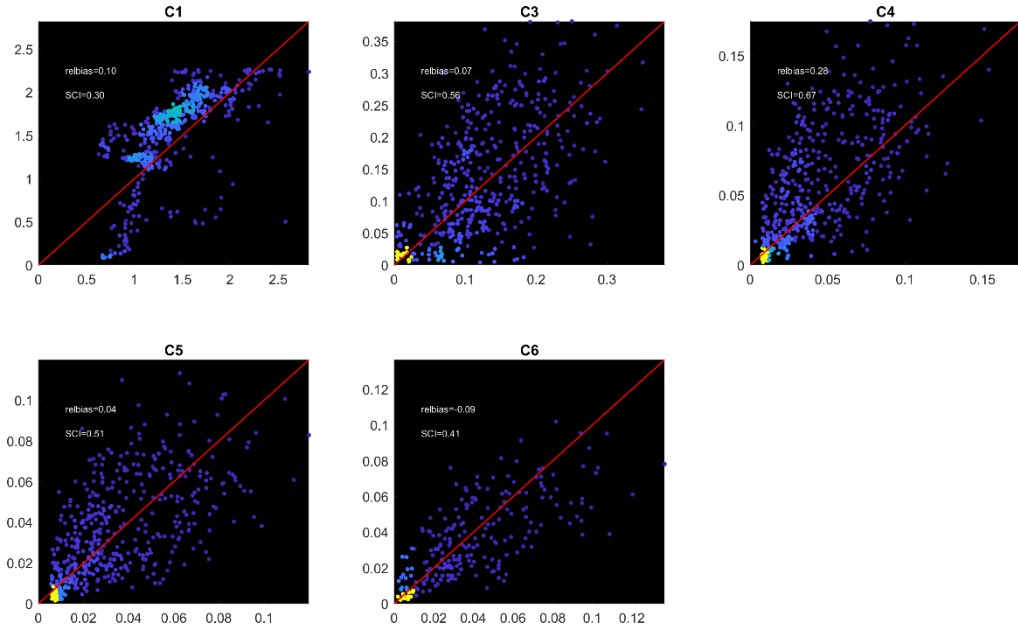

**Figure 27 Scatterplots (heat maps) of computed vs. observed Hm0 wave heights. Large-scale model. For color legend refer to Figure 11.**

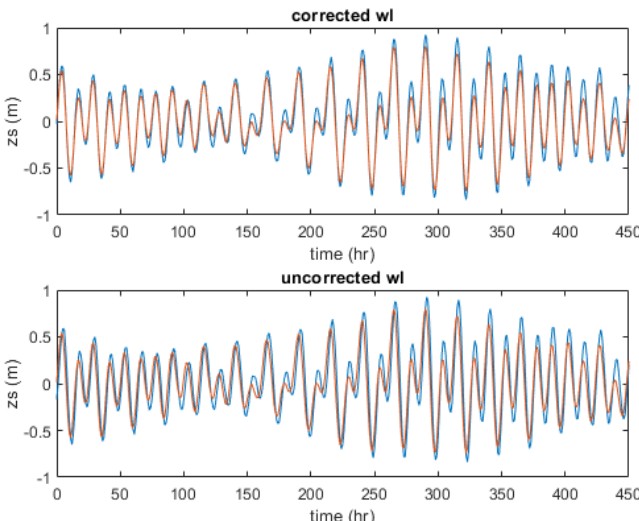

**Figure 28 GTSM hindcast of water level (blue) vs. observed (red). Lower panel: uncorrected except for 8-hr shift from GMT to local time; top panel: GTSM model shifted by one hour to GMT+7hr.**

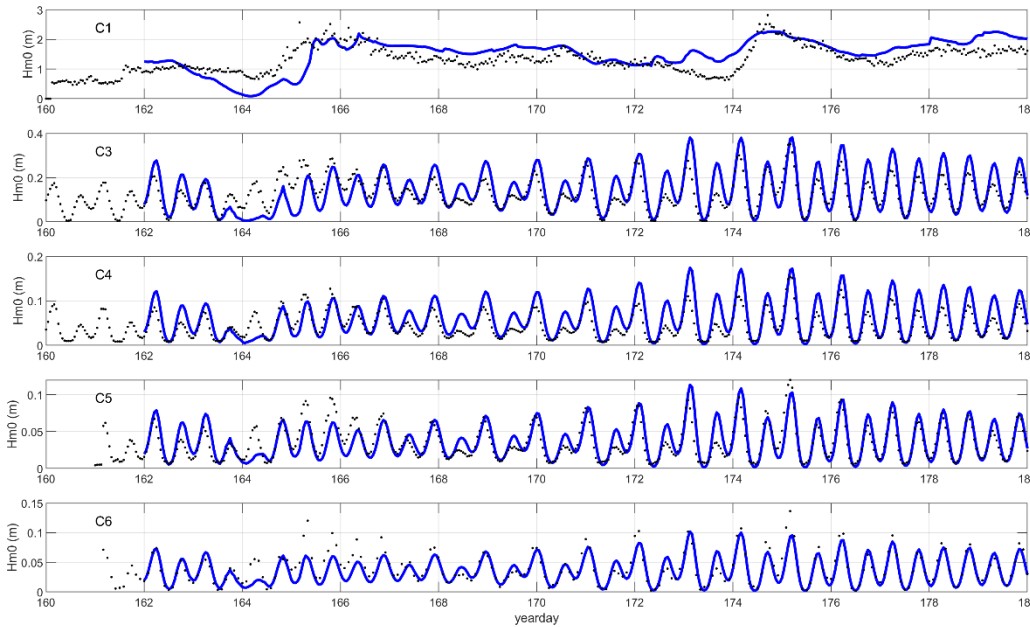

**Figure 29 Time series of Hm0 wave height across the reef at Ningaloo; observations (black dots) against model simulation (blue drawn lines). Large-scale model, simulation results shifted by one hour.**

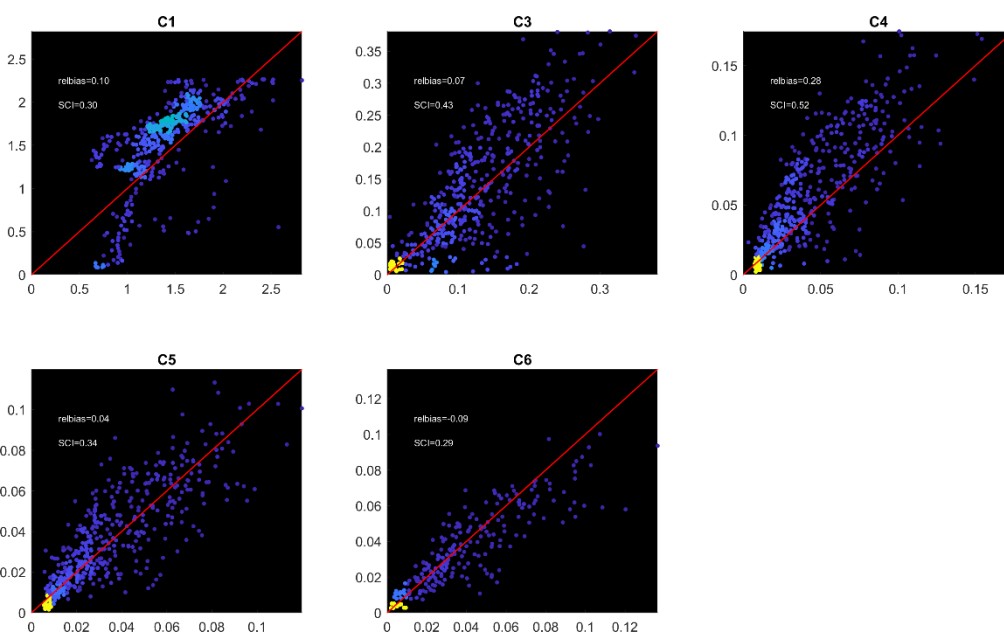

**Figure 30 Scatterplots (heat maps) of computed vs. observed Hm0 wave heights. Large-scale model. Simulation results shifted by one hour. For color legend refer to Figure 11.**

**Effect of wave setup on wave propagation and decay.**

For a given water level, the local wave setup can have an important influence on the wave decay. In such cases, a coupled hydrodynamic model is needed to provide this non-uniform water level. We tested the effect of wave setup by using Delft3D-FM with the in-built SnapWave solver to check how important this effect is. The model was set up for the local grid and fed with constant water level boundary conditions, over the same period as for the other simulations. In Figure 31 the wave heights for the simulation without wave setup are compared with those including the effect of wave setup. The effect is significant, in the order of 2-3 cm or about 10-20% of the local wave height. Still, in light of the uncertainty in the local water level, this is still a relatively minor effect.

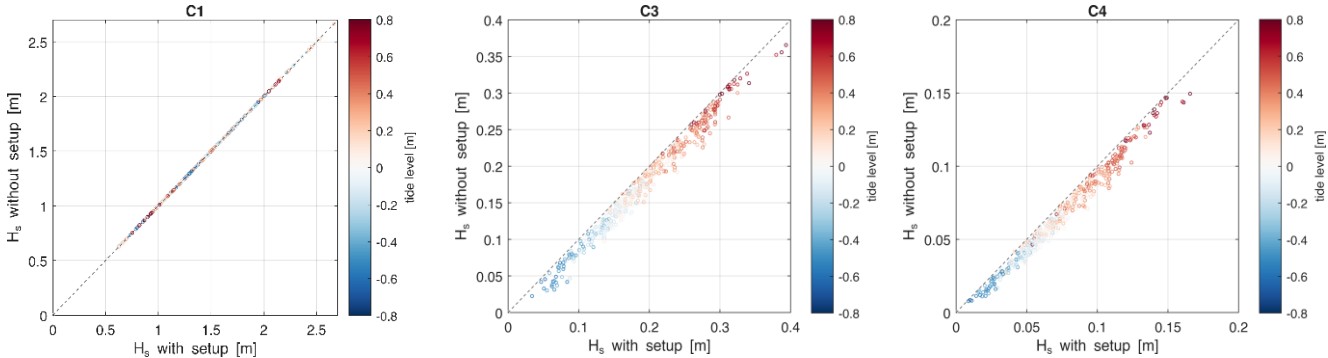

**Figure 31 Comparison of coupled Delft3D-FM - SnapWave model with and without setup, for observation points 1, 3 and 4.**

**4.5 Haringvliet intercomparison with unSWAN**

The Haringvliet mouth is a shallow area seaward of a closed-off estuary, with sand banks, shoals and channels and has long been used as a test case for the models HISWA, SWAN and recently unSWAN. The model can be readily downloaded from the SWAN website https://swanmodel.sourceforge.io/download/download.htm and was used here to check the difference in results and performance between SnapWave and unSWAN, on the exact same triangular grid. Two modifications were made to the unSWAN input:

- instead of a 1D spectrum, a similar parametric JONSWAP spectrum was imposed, with peak period Tp=8s, wave height Hm0=3.2m and mean direction of 270°N; peak enhancement factor was 3.3.
- the bed friction was set to 'COLLins' with value of 0.02, to facilitate comparison with SnapWave, fw=0.02.
- wind was turned off in both cases.

For SnapWave, the unSWAN grid files were converted to a NetCDF UGrid file. In both cases, the same convergence criterion of 0.02 was used. SnapWave converged for 100% of grid nodes in 3 iterations, which took 0.03 s, whereas unSWAN took 5 iterations to converge for 99.7% of the grid nodes, in approximately 6 s, a factor of 200 slower.

The results are shown in Figure 32; in most of the area, the pattern of Hm0 wave height is very similar. We present the point-by-point comparison in Figure 33, with a low relative bias of 0%, a scatter index of 9% and a skill of 0.98.

We may conclude that in this kind of nearshore application dominated by refraction, shoaling, bed friction and wave breaking dissipation, differences between SnapWave and unSWAN are small in the light of other uncertainties, and that SnapWave is two orders of magnitude faster.

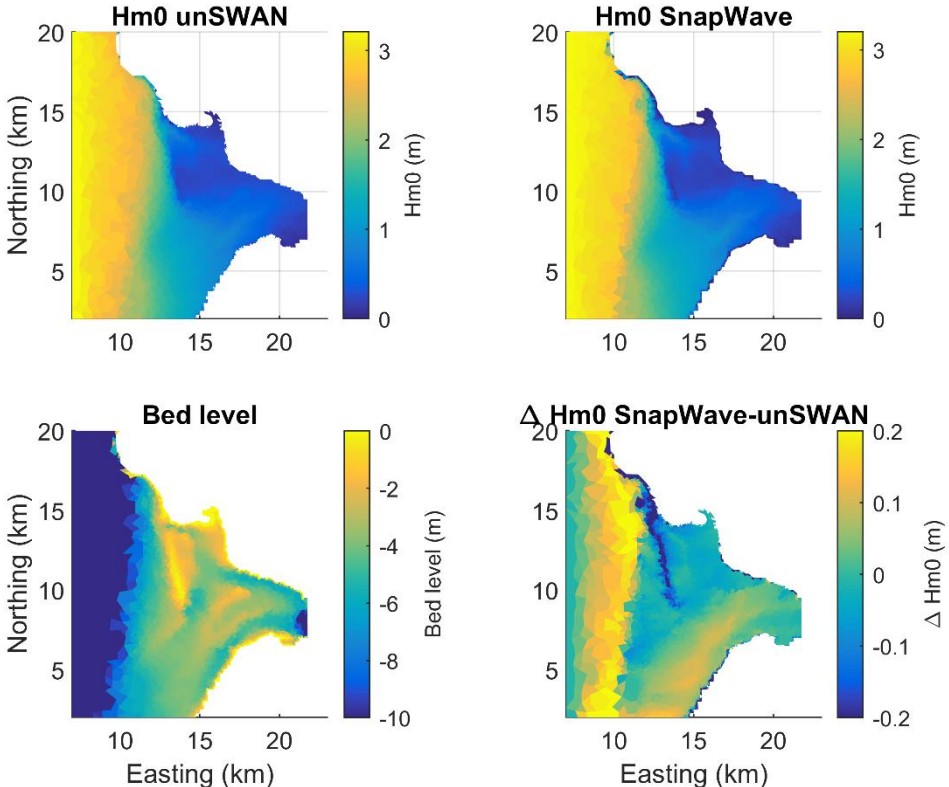

**Figure 32 Comparison of unSWAN and SnapWave for the Haringvliet case. Top left and top right panels: Hm0 computed by unSWAN resp. SnapWave; lower left panel: bed level; lower right panel: difference between the models.**

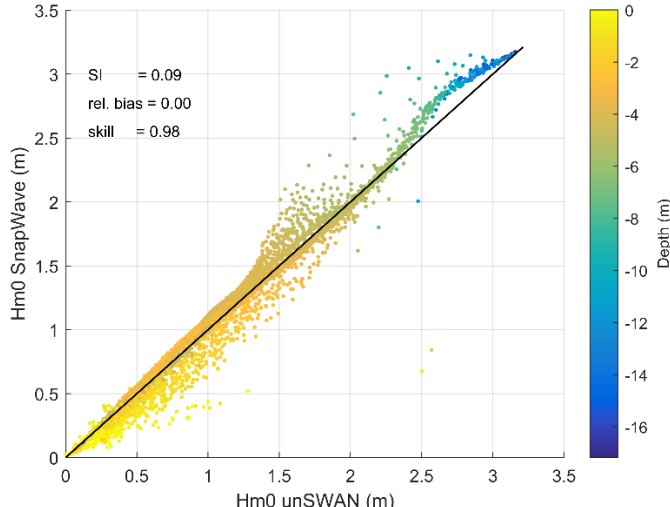

**Figure 33 Scatterplot of Hm0 SnapWave against Hm0 unSWAN, for rectangular area indicated in Figure 32**

## 5. Discussion

### 5.1 Model features

The SnapWave model uses unstructured grids based on the NetCDF ugrid convention (https://ugrid-conventions.github.io/ugrid-conventions/), which can consist of a combination of triangular or quadrangular cells, for which no particular restrictions apply. The numerical method converges quickly to high accuracy (typically a relative error of 10e-5 within 10 iterations or less), and in closed-coast cases the first sweep of the first iteration resolves most of the final solution. For the linear shoaling and refraction case we show that the model results between rectilinear and unstructured meshes are quite comparable and that the having the right resolution where the steepest gradients are governs the accuracy. In more general cases, such as determining wave fields around an island, the model allows omnidirectional propagation and refraction. The boundary conditions can be a combination of Dirichlet or Neumann conditions. The model provides convenient NetCDF output using CF conventions.

### 5.2 Model behaviour on open coasts and islands

In the schematic verification cases SnapWave accurately reproduces linear theory for longshore uniform coasts. For the case of a flat circular reef the model produces qualitatively similar results as the analytical solution by (Mandlier and Kench, 2012), with a fair match in wave height patterns, showing focusing towards the same area leeward of the shoal centre. For the case of a circular island we illustrate the rapid convergence around it and the smooth and realistic wave height pattern.

## 5.3 Model efficiency

All simulations were carried out on a HP ZBook Studio 16 inch G10 Mobile Workstation PC, with 13th Gen Intel(R) Core(TM) i7-13800H, 2500 Mhz, 14 Core(s), 20 Logical Processor(s) and 64GB RAM.

The large-scale COAST3D model has approximately 340,000 net nodes, as it covers the Dutch coastal zone excluding the Wadden Sea, at a resolution down to 100m. The extra refinements near the Coast3D site only added relatively few extra points,

and the higher resolution does not influence the implicit solution in any way. The computation of one wave field took 1.9 s on average; for the 12 days of the simulation at hourly intervals this took 9 minutes.

The local model has approximately 5,000 nodes; computation of one wave field took 16 ms, and the 12-day period at half-hour intervals took 9 seconds.

Per node, directional bin and condition, the large-scale model took 0.31 microseconds, whereas the local model only needed

0.17 microseconds. This difference can be attributed to the fact that the more complex large-scale model typically took 10 iterations to fully converge, where the local model typically took 6.

In Table 3 the run times and model characteristics are shown for all field validation cases. In general we can conclude that the model takes around 0.15 microseconds per node, directional bin and wave condition for the simplest rectangular grids, and around 0.3 microseconds for more complex, unstructured grids. The St Croix model is an outlier with 0.6 microseconds, which

cannot be explained by its convergence characteristics, which are very similar to e.g. the Ningaloo large-scale grid.

The Haringvliet case compares the performance between SnapWave and unSWAN, and shows that on the exact same horizontal grid, the SnapWave model is two orders of magnitude faster while giving, for this shallow nearshore case with complex bathymetry, very comparable results.

**Table 3 Overview of run time characteristics for all field validation models**

| Model | # nodes | # wave bins | # wave conditions | time (s) | time/condition (s) | time/condition/node/bin (microseconds) |
|---|---|---|---|---|---|---|
| Coast3D large-scale | 338292 | 18 | 288 | 540 | 1.9 | 0.31 |
| Coast3D local | 4964 | 18 | 576 | 9 | 0.016 | 0.17 |
| Ameland large-scale | 226258 | 36 | 1440 | 3450 | 2.4 | 0.3 |
| St Croix | 36236 | 36 | 3672 | 3000 | 0.82 | 0.6 |
| Ningaloo large-scale | 100146 | 36 | 456 | 480 | 1.1 | 0.3 |
| Ningaloo local | 46146 | 36 | 456 | 120 | 0.26 | 0.16 |

| Haringvliet triangular mesh | 5961 | 36 | 1 | 0.03 | 0.03 | 0.14 |
|---|---|---|---|---|---|---|
| Haringvliet triangular mesh, unSWAN | 5961 | 36 | 1 | 6 | 6 | 28 |

## 5.4 Method to transform wave conditions from ERA5 to nearshore

ERA5 performed well in all cases, with absolute bias typically less than 10%, scatter index 20-25%. Extreme events may be underestimated where ERA5 cannot resolve the atmospheric scale of the depression, as in the case of the US Virgin Islands. Results for nearshore locations have similar relative bias (~10%) and somewhat higher scatter index (~30%). The case of Ningaloo Reef poses a severe challenge because of the high friction losses, represented by a uniform friction coefficient of 0.6, and because of its sensitivity to the water level, where even a small phase error leads to large deviations in water level and hence shallow water wave heights. In this case the averaged scatter index for points on the reef is around 40% for the large model forced by ERA5, against around 30% for the purely local model. The effect of neglecting wave setup in the case of Ningaloo Reef was significant but small compared to this scatter. As this is the case study that is most sensitive to wave setup, we may conclude in general that neglecting wave setup in predicting nearshore wave conditions is acceptable in view of other uncertainties.

## 5.5 Limitations

The SnapWave model considers directionally spread waves with a single representative frequency, which introduces errors for multi-peaked spectra. Particularly the use of the peak frequency as the characteristic frequency can introduce large fluctuations in the modelled wave celerity and group speed, if swell and local wind waves compete for dominance of the spectrum. The use of a characteristic period $Tm10$ that best represents the mean group velocity is then recommended:

$$T_{m-1,0} = \frac{\int S f^{-1} df}{\int S df} \qquad (19)$$

Where S is the spectral density (m²/Hz) and $f$ is the frequency (Hz)There is a wide range of bi- or multi-model wave conditions that could occur and the current model does not represent this well; this is likely one of the causes for the scatter in the nearshore comparisons against data. However, we see possible improvements in future versions to better deal with this.

First, we could improve the input functionality by reading in 2D spectra and converting these to 1D directional spectra and the $Tm10$ wave period. This would already improve the prediction for systems with spectral partitions from different directions but similar frequencies. Alternatively, such 1D directional spectra could be generated from integral parameters of the sea and swell bands. Both are relatively minor implementation issues that would not affect the overall solution method.

Second, we could apply different characteristic frequencies per directional bin. This would allow us to represent swell and sea from different directions and with clearly different frequencies. This is worth investigating but less straightforward, particularly in combination with wind growth.

The functionality described here does not include wave growth by wind, although this process has been implemented and is currently being tested. The model is stationary and is therefore suited for swell propagation and wave propagation over limited distances, as is typically the dominant situation in coastal areas. It can provide a fast alternative to more complex models such as SWAN when the dominant processes are wave shoaling, refraction and dissipation by friction and depth-limited wave breaking.

## 6. Conclusions

The SnapWave model presented here provides an efficient way to propagate wave conditions from the ERA5 hindcast, or similar global wave hind- or forecasts, to the nearshore. We have shown that the model correctly simulates nearshore wave propagation and dissipation for directionally spread waves specified at points typically 50-100 km offshore. For the cases we tested the ERA5 hindcast provides adequate boundary conditions and the combination of ERA5 and SnapWave is able to reproduce time series of wave heights at nearshore locations with significant skill. Although we have only tested the method in a few locations, we believe this approach can be used on open coasts anywhere and has the potential to be used as part of large-scale to global assessments that rely on nearshore wave conditions.

**Code and data availability**

The current version of the model, as well as the input, data and test scripts for the test cases are available from the project website: https://github.com/danoroelvink/snapwave/ under the licence GNU Lesser General Public License as published by the Free Software Foundation version 2.1 or higher of the License. The exact version of the model used to produce the results used in this paper is archived on Zenodo https://doi.org/10.5281/zenodo.14831094 (Roelvink et al., 2025).

**CRediT Author contribution statement**

**DR:** Conceptualization, methodology, software, validation, writing – original draft; **MvO**: methodology, software; **JR**: software, validation, writing – review & editing; **MvdL**: software, validation, writing – review & editing.

**Competing interests**

The authors declare that they have no conflict of interest.

## Acknowledgements

The work presented here was funded as part of the FHICS (Forecasting Hurricane Impacts on CoastS) project, funded by the US Office of Naval Research and the ShorelineS-TKI project as part of the Delta Technology Topsector Knowledge Innovation. Data for the Ningaloo Reef case was provided by Prof. Ryan Lowe of the University of Western Australia.

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

## Appendix A Error metrics

**Table A.1 Definitions of error metrics**

| Parameter | Formula (m=measured; c=computed) | Description |
|---|---|---|
| Pearson correlation rho | $\dfrac{Cov(m,c)}{\sigma_m \sigma_c}$ | Correlation coefficient, indicating strength of a linear relationship between random variables m and c |
| Scatter Index SCI | $\dfrac{\sqrt{\overline{(c-m)^2}}}{\max\left(\sqrt{\overline{m^2}}, \left|\overline{m}\right|\right)}$ | Relative measure of the scatter between model and data. The error is normalised with the maximum of the rms of the data and the absolute value of the mean of the data; this avoids strange results for data with small mean and large variability |
| Relative bias | $\dfrac{\overline{c-m}}{\max\left(\sqrt{\overline{m^2}}, \left|\overline{m}\right|\right)}$ | This is a relative measure of the bias, normalised in the same way as the Scatter Index |
| Brier skill | $1 - \dfrac{var(c-m)}{var(m)}$ | This parameter relates the variance of the difference between data and model to the variance of the data. skill=1 means perfect skill; skill=0 means no skill; skill¡0 means result is worse than doing nothing. |

**Table A.2 Error metrics shoaling and refraction test, Hm0**

| runid | dir | rho | sci | relbias | skill |
|---|---|---|---|---|---|
| uniform_20 | 0 | 0.993 | 0.002 | -0.001 | 1.000 |
| uniform_10 | 0 | 0.994 | 0.001 | 0.000 | 1.000 |
| variable_40_10 | 0 | 0.994 | 0.001 | 0.000 | 1.000 |
| uniform_20 | 30 | 0.990 | 0.004 | -0.002 | 1.000 |
| uniform_10 | 30 | 0.994 | 0.003 | -0.002 | 1.000 |
| variable_40_10 | 30 | 0.993 | 0.003 | -0.003 | 1.000 |
| uniform_20 | 45 | 0.991 | 0.007 | -0.005 | 1.000 |
| uniform_10 | 45 | 0.993 | 0.006 | -0.005 | 1.000 |
| variable_40_10 | 45 | 0.993 | 0.007 | -0.006 | 1.000 |

**Table A.3 Error metrics shoaling and refraction test, wave direction**

| runid | dir | rho | sci | relbias | skill |
|---|---|---|---|---|---|
| uniform_20 | 30 | 0.992 | 0.015 | -0.009 | 1.000 |
| uniform_10 | 30 | 0.993 | 0.011 | -0.009 | 1.000 |
| variable_40_10 | 30 | 0.993 | 0.013 | -0.010 | 1.000 |
| uniform_20 | 45 | 0.991 | 0.017 | -0.010 | 1.000 |
| uniform_10 | 45 | 0.993 | 0.013 | -0.011 | 1.000 |
| variable_40_10 | 45 | 0.993 | 0.014 | -0.012 | 1.000 |

**Table A.4 Error statistics Coast3D local model, gamma=0.70**

| point | rho | sci | relbias | skill |
|---|---|---|---|---|
| 2 | 0.9028 | 0.229 | 0.181 | 0.948 |
| 1a | 0.9598 | 0.132 | 0.077 | 0.983 |
| 1b | 0.9535 | 0.163 | 0.120 | 0.974 |
| 1c | 0.9358 | 0.16 | 0.097 | 0.974 |
| 1d | 0.9334 | 0.183 | 0.133 | 0.966 |

**Table A.5 Error statistics Coast3D large-scale model, gamma=0.7**

| point | rho | sci | relbias | skill |
|---|---|---|---|---|
| 8 | 0.883 | 0.216 | -0.044 | 0.953 |
| 2 | 0.1813 | 0.434 | 0.064 | 0.812 |
| 1a | 0.6003 | 0.304 | 0.083 | 0.907 |
| 1b | 0.7143 | 0.289 | 0.120 | 0.916 |
| 1c | 0.7318 | 0.273 | 0.098 | 0.925 |
| 1d | 0.7313 | 0.291 | 0.139 | 0.915 |

**Table A.6 Error statistics Coast3D large-scale model, gamma=0.75**

| point | rho | sci | relbias | skill |
|---|---|---|---|---|
| 8 | 0.883 | 0.216 | -0.044 | 0.953 |
| 2 | 0.1935 | 0.441 | 0.078 | 0.806 |
| 1a | 0.6104 | 0.318 | 0.120 | 0.899 |
| 1b | 0.7142 | 0.316 | 0.167 | 0.900 |
| 1c | 0.7323 | 0.297 | 0.146 | 0.912 |
| 1d | 0.7296 | 0.323 | 0.190 | 0.895 |

**Table A.7 Error statistics Ameland Inlet, Hm0**

| point | rho | sci | relbias | skill |
|-------|-----|-----|---------|-------|
| AZB11 | 0.924 | 0.236 | -0.131 | 0.944 |
| AZB12 | 0.949 | 0.197 | -0.100 | 0.961 |
| AZB21 | 0.808 | 0.300 | -0.171 | 0.910 |
| AZB22 | 0.789 | 0.266 | -0.064 | 0.929 |
| AZB31 | 0.841 | 0.291 | -0.152 | 0.915 |
| AZB32 | 0.792 | 0.304 | -0.072 | 0.908 |

**Table A.8 Error statistics Ameland Inlet, Tp**

| point | rho | sci | relbias | skill |
|-------|-----|-----|---------|-------|
| AZB11 | 0.819 | 0.152 | -0.021 | 0.977 |
| AZB12 | 0.787 | 0.157 | -0.014 | 0.975 |
| AZB21 | 0.592 | 0.216 | -0.022 | 0.953 |
| AZB22 | 0.490 | 0.236 | -0.011 | 0.944 |
| AZB31 | 0.339 | 0.268 | -0.061 | 0.928 |
| AZB32 | 0.446 | 0.289 | 0.039 | 0.917 |

**Table A.9 Error statistics St Croix, Hm0**

| point | rho | sci | relbias | skill |
|-------|-----|-----|---------|-------|
| christiansted | 0.849 | 0.210 | -0.037 | 0.956 |
| fareham | 0.911 | 0.170 | -0.076 | 0.971 |

**Table A.10 Error statistics Ningaloo, local model**

| point | rho | sci | relbias | skill |
|-------|-----|-----|---------|-------|
| C1 | 0.990 | 0.046 | 0.022 | 0.998 |
| C3 | 0.897 | 0.236 | 0.117 | 0.944 |
| C4 | 0.908 | 0.337 | 0.248 | 0.887 |
| C5 | 0.907 | 0.251 | -0.005 | 0.937 |
| C6 | 0.904 | 0.335 | -0.143 | 0.888 |

**Table A.11 Error statistics Ningaloo, large-scale model, uncorrected water levels**

| point | rho | sci | relbias | skill |
|-------|-----|-----|---------|-------|
| C1 | 0.643 | 0.285 | 0.094 | 0.919 |
| C3 | 0.549 | 0.533 | 0.071 | 0.716 |
| C4 | 0.607 | 0.651 | 0.271 | 0.577 |
| C5 | 0.598 | 0.500 | 0.035 | 0.750 |

| point | rho | sci | relbias | skill |
|-------|-----|-----|---------|-------|
| C6 | 0.697 | 0.407 | -0.085 | 0.834 |

**Table A.12 Error statistics Ningaloo, large-scale model, corrected water levels**

| point | rho | sci | relbias | skill |
|-------|-------|-------|---------|-------|
| C1 | 0.649 | 0.284 | 0.096 | 0.919 |
| C3 | 0.750 | 0.411 | 0.072 | 0.831 |
| C4 | 0.820 | 0.502 | 0.272 | 0.748 |
| C5 | 0.819 | 0.336 | 0.036 | 0.887 |
| C6 | 0.862 | 0.288 | -0.084 | 0.917 |

**Appendix B Parameterization of frequency spectrum**

