# Peer review of "SnapWave: fast, implicit wave transformation from offshore to nearshore"

_EGUsphere, 2025_

## Author Response (AR1)

Review of "SnapWave: fast, implicit wave transformation from offshore to nearshore" by Roelvink et al., manuscript egusphere-2025-492 (my first review of this manuscript).

1. *[Comment]* The authors present a new wave model specifically for transforming offshore waves to the coast. The paper is somewhat long due to the significant number of practical test cases. However, there is value in all cases, and the combination of test cases comprehensively shows the capability of the model. The material is suitable for publication but needs a bit of work. *[Response]* We thank the reviewer for the careful and positive review. We appreciate the overall assessment and agree that there was room for improvement and clarification.

2. *[Comment]* **Introduction**: The references to previous work in the introduction are appropriate, but a little sparse. In particular Bill O'Reilly's work with several wave models for transforming offshore conditions to the coast for CDIP on the US West Coast is relevant. Moreover, Aaron Roland's work with triangular grids with his own wave model and with WW3 and Jose-Henrique Alves' implementation of the latter WW3 option at NOAA for the Great Lakes is relevant here. Aaron's work contradicts the authors; statements on lines 33-35. *[Response]* We agree and have rewritten the introduction to better represent earlier work and to have a more logical flow leading to the need for this new model.

3. *[Comment]* Note that the format of the references is all over the place. Please standardize. *[Response] Something had gone wrong there, we hope the references are now in the right and consistent format.*

4. *[Comment]* Note moreover, that WW3 should be referred to using peer-reviewed papers, not a report and a manual (e.g., 1991 JPO reference for first WAVEWATCH, 2002 W&F first reference to WW3). *[Response]* Agreed, done.

5. **Model description**: This section is inadequate. First, it is claimed that the model solves the action equation (line 52), yet the paper only deals with an energy equation. In the energy equation, the spectrum *ee* is a one-dimensional spectrum according to the introduction, but that is not confirmed here. Is the spectral frequency / period invariant here? The linkage between *ee* and E is not defined, and there is no description here on which source terms are used explicitly (only depth induced breaking?) and if the other source terms are included at all (I believe HISWA did have parametric wave growth included, but I may be wrong on that account). *[Response]* The model description has been extended and rewritten. It now starts with the description of the combined action and energy balance, including wind growth terms in section 2.1, after which we argue, in section 2.2, that for many applications wind growth and current refraction can be neglected so the equations reduce to the wave energy balance. Wind growth is in the model but will be described and validated in an upcoming paper.

6. *[Comment]* Line 55 mentions "the" unstructured grid without defining it. The grid was "introduced" in Fig. 1, but I did not start understanding it until the practical grid examples were given. I started understanding it better with the reference to the NeCDF standard on line 455. This should be in the introduction or in Section 2! As SnapWave effectively uses a stepwise increased resolution, the authors should refer to the SMC grid by Jan-Guo Li and the quadtree approach by Stephane Popinet. I also would love to know if the square cells need to be (quasi-) orthogonal and/or can be curvilinear, and how energy conservation is addressed in the numerical scheme. *[Response]* We now dedicate a section 2.3 to describing the numerical grid.

7. *[Comment]* Please elaborate on sweeping mentioned on line 57, and on the difference between the sweeping and the iterative solution. *[Response]* Section 2.4 on the

discretization and solution method is now much more detailed and describes the sweeping and iteration process more clearly, we hope.

8. *[Comment]* Line 93 "Obviously". Please elaborate for those of us more familiar with explicit schemes. *[Response]* this is now covered in the explanation of the sweeping and iteration process.

9. *[Comment]* **Verification**: Section 3.1 verifies refraction, shoaling and breaking, not just the first two processes. This section is also a little sloppy as it claims on lines 114-117 that two = one + one + one, and that there are two cross-shore resolutions. *[Response]* This test only includes wave refraction and shoaling and no breaking, to allow comparison with the analytical solution where wave energy is conserved (cross-shore wave energy flux is constant). Indeed, we have three grid configurations, this has been corrected.

10. *[Comment]* **Field validation:** Please provide legends for the scatter plot colors. *[Response]* Colors indicate relative point density, scaled by maximum density of data points. A colorbar with label is given in Figure 11 and referred to in all other scatterplots where the points are colored by relative density.

11. *[Comment]* Out of curiosity, how do you deal with the ever-changing bathymetry for depths less than 10 fathom on the Dutch coast*? [Response]* for the Coast3D and Ameland case we updated the local bathymetry based on the most recent observations in both cases. In general, we would try to use the most appropriate sounding available. When in use as part of a morphodynamic model, the bathymetry is continuously updated by the system.

**Citation**: https://doi.org/10.5194/egusphere-2025-492-RC1

*[Comment]* This manuscript introduces SnapWave, a new, efficient, implicit, unstructured-grid wave propagation model designed for transforming offshore wave conditions to nearshore environments. The model focuses on essential processes like refraction, shoaling, bottom friction, and wave breaking, while omitting less critical aspects such as full spectral wind-wave growth to achieve computational speed. It is positioned as a tool for integration with other coastal models (e.g., ShorelineS, XBeach, Delft3D-FM, SFINCS) and demonstrates its applicability through verification against analytical solutions and field validations in diverse coastal settings.

The paper is well-structured, with clear descriptions of the model's numerical method, discretization, and solution scheme. It aligns well with GMD's scope, which emphasizes the development, evaluation, and application of geoscientific models. The emphasis on efficiency and robustness for global open-coast applications is a valuable contribution, particularly in the context of climate change-driven coastal risk assessments using datasets like ERA5.

However, there are areas for improvement in clarity, completeness, and documentation, which could strengthen the manuscript. Overall, the work is sound and innovative, but minor revisions are needed to address these issues. *[Response]* We thank the reviewer for the positive comments and careful review, and the helpful suggestions for improvement of the manuscript.

Comments:

1. *[Comment]* The manuscript describes a fast, implicit, unstructured solver for stationary wave propagation. Similar capabilities exist in unSWAN and STWAVE. The authors should delineate the conceptual and algorithmic differences (for example, the back-tracing and multi-sweep strategy, as well as the discretization choices in Eqs. 2.6–2.7), and they should provide at least one side-by-side benchmark against SWAN or STWAVE on a shared domain, reporting both accuracy and computational cost. *[Response]* We have explained the numerical method including the sweeping and iteration process in a much more complete way; though the principles are very similar, the main differences with STWAVE and unSWAN are in the numerical grid, which can be a combination of quadrilaterals and triangles instead of STWAVE's rectangular grids and unSWAN's purely triangular grids. As requested by the reviewer, we have used one of unSWAN's test cases, the Haringvliet model, to carry out a one-on-one comparison between unSWAN and SnapWave, as outlined in section 4.5. Here we show that for such a shallow, complex area, both models provide very similar wave fields, where SnapWave is approx.. 200 times faster.

2. *[Comment]* The discretization (Eq. 2.6) and tridiagonal solve (Eq. 2.7) are outlined, but important implementation details remain unclear. Specifically, the interpolation weights and stencil construction for the upwind point "u," the treatment of wetting/drying or very shallow layers on reefs, the precise definition and calibration of the dissipation coefficient $\alpha$ in Eq. (2.2), the handling of open and lateral boundary conditions, and the ordering of directional sweeps relative to the dominant direction all need to be described. The choice of convergence tolerance ($10^{-5}$) should also be justified. Providing pseudo-code or an algorithmic summary with sensitivity plots would significantly improve reproducibility. *[Response]* The descriptions of the numerical grid and solution method have been substantially extended, and we hope that sections 2.3 and 2.4 now explain all necessary details in a way that allows reproduction.

3. *[Comment]* The single-frequency approximation based on the peak period is a key modeling assumption, but the manuscript does not quantify its limitations. The authors should

demonstrate the impact of this assumption with a synthetic bi-modal spectrum test compared against a spectral model, and they should analyze a field case where bimodal spectra are present. Sensitivity to the choice of representative frequency (peak, maximum energy, or moment-based) should also be reported. *[Response]* We appreciate the importance of this limitation and have devoted much more text to it under section 5.5 Limitations. We feel, however, that a single test of bimodal spectra vs single-peak spectra would not provide enough information to extract meaningful guidance on this. Rather, we'd like to explore the two options indicated that could improve the representation of bimodal spectra in a separate paper, as this would expand the scope of the current paper too much.

4. *[Comment]* The Ningaloo experiments illustrate that wave setup substantially modifies effective depths and hence wave transformation. Because the model is positioned as a nearshore transformer suitable for reef environments, the manuscript should either demonstrate coupling with a simple setup model or explicitly quantify the expected errors when setup is not included. *[Response]* We thank the reviewer for the suggestion and have run a coupled simulation with and without the effect of setup. The effect on the first two points on the reef is small but relevant and we discuss it at the end of the section on Ningaloo reef.

5. *[Comment]* To satisfy the standards of GMD, the open-source archive must provide complete case input data (grids, bathymetry, wave and tide forcings, and parameter files) together with scripted workflows to regenerate all figures and tables. A concise "How to reproduce" section should list the DOI, software environment, compiler information, and run commands for each case. *[Response]* We have included all the relevant input, run scripts and plot scripts in the repository under *testcases*.

**Citation**: https://doi.org/10.5194/egusphere-2025-492-RC2